# DT-BEHRT: Disease Trajectory-aware Transformer for Interpretable Patient Representation Learning

## Abstract

The growing adoption of electronic health record (EHR) systems has provided unprecedented opportunities for predictive modeling to guide clinical decision making. Structured EHRs contain longitudinal observations of patients across hospital visits, where each visit is represented by a set of medical codes. While sequence-based, graph-based, and graph-enhanced sequence approaches have been developed to capture rich code interactions over time or within the same visits, they often overlook the inherent heterogeneous roles of medical codes arising from distinct clinical characteristics and contexts. To this end, in this study we propose the Disease Trajectory-aware Transformer for EHR (DT-BEHRT), a graph-enhanced sequential architecture that disentangles disease trajectories by explicitly modeling diagnosis-centric interactions within organ systems and capturing asynchronous progression patterns. To further enhance the representation robustness, we design a tailored pre-training methodology that combines trajectory-level code masking with ontology-informed ancestor prediction, promoting semantic alignment across multiple modeling modules. Extensive experiments on multiple benchmark datasets demonstrate that DT-BEHRT achieves strong predictive performance and provides interpretable patient representations that align with clinicians' disease-centered reasoning.

## 1 Introduction

With the rapid growth of electronic health record (EHR) data, predictive modeling has become an important tool for generating actionable insights to support clinical decision making. Structured EHRs consist of trajectories of hospital visits, where each visit contains a collection of various medical codes that capture patients' diagnoses, medications, procedures, and laboratory tests. Sequence modeling has therefore become a prominent approach in EHR-based predictive analysis. Studies such as BEHRT (Li et al., 2020), Med-BERT (Rasmy et al., 2021), and ExBEHRT (Rupp et al., 2023) adapted the BERT (Devlin et al., 2019) framework and pre-trained models on structured EHR datasets of varying sizes. However, existing sequence-based methods generally face two key challenges when dealing with multiple codes within the same visits: (1) the order of codes is often unreliable since they are reported by coding practices rather than true clinical chronology, and (2) code co-occurrence and dependencies are often inadequately captured when visits are represented as multi-hot vectors. These challenges have motivated the development of graph-based approaches, such as homogeneous (Song et al., 2023), heterogeneous (Chen et al., 2024), and hypergraph (Xu et al., 2023) models that aim to explicitly leverage structural relationships in EHR data. However, graph-based methods often struggle to capture sequential dependencies across visits.

Graph-enhanced sequence approaches have therefore been proposed to integrate the strengths of both paradigms. G-BERT (Shang et al., 2019) incorporates a graph to enrich medical code embeddings with hierarchical ontology structures. GCT (Choi et al., 2020) was among the first to model intra-visit code relationships with graphs, while TPGT (Hadizadeh Moghaddam et al., 2025) and DeepJ (Li et al., 2025) strengthened temporal modeling capabilities. HEART (Huang et al., 2024) connects multiple visit representations of the same patient into a graph to enable message passing across visits. A more detailed overview of related work is provided in the Appendix A. However,

existing models largely overlook the fact that different types of medical codes play fundamentally distinct roles in shaping a patient's health representation.

Medical codes are inherently heterogeneous, reflecting their diverse clinical roles and characteristics. For example, procedures and medications often reflect treatment pathways, therefore are inherently temporal related over time but exhibit limited interactions within a single visit. In contrast, diagnosis codes serve as the driving force in shaping a patient's health trajectory. They are more interactive, with dense connections to other diseases within the same organ system, and also facilitate influence across different systems over time. These differences highlight the need for a code-type-specific algorithmic paradigm, supported by specialized modeling modules that explicitly account for the distinct roles of different code categories.

In this study, we introduce the Disease Trajectory-aware Transformer for EHR (DT-BEHRT), which directly addresses the aforementioned gaps. Unlike homogeneous modeling approaches that treat all codes uniformly, DT-BEHRT incorporates tailored modules to capture the fundamental differences between diagnosis and treatment codes. By explicitly encoding disease trajectories and corresponding treatment pathways, our framework models both the temporal dynamics and system-wise interactions of diagnoses across visits. This design is essential, as many downstream clinical prediction tasks, such as mortality prediction and disease phenotyping, are inherently dependent on rich representations of disease progression. Our key contributions are threefold:

- **Model architecture.** We introduce DT-BEHRT, a novel graph-enhanced sequence model that models and interprets longitudinal EHR by leveraging diagnosis-centric interactions in organ systems, and formulating personalized disease progression patterns for patient representation learning.

- **Pre-training framework.** We design a tailored pre-training framework that combines a novel masked code prediction task with ancestor code prediction. This objective enhances module alignment across functional components and consistently improves the robustness of patient representations.

- **Comprehensive evaluation.** We conduct extensive experiments across diverse clinical prediction tasks, where DT-BEHRT achieves competitive performance and maintains robustness across subgroups. Through case studies, we further demonstrate that its design aligns with clinicians' diagnostic reasoning, providing both accuracy and interpretability.

## 2 PRELIMINARY

In this section, we introduce key concepts and notations that are essential for introducing our method. In EHR data, each medical event $c$ in a patient's clinical trajectory is recorded as a code drawn from a vocabulary of unique medical codes, denoted as $\mathcal{C} = \{c_1, c_2, \ldots, c_{|\mathcal{C}|}\}$, where $|\mathcal{C}|$ denotes the total size of the vocabulary. Meanwhile, each code can be categorized into one of four medical event categories: diagnosis ($\mathcal{D}$), medication ($\mathcal{M}$), laboratory test ($\mathcal{L}$), or procedure ($\mathcal{P}$). Formally, the vocabulary can be expressed as the union $\mathcal{C} = \mathcal{D} \cup \mathcal{M} \cup \mathcal{L} \cup \mathcal{P}$. Based on these notations, a patient's clinical trajectory can be naturally modeled as a sequence of temporally ordered hospital visits, denoted as $\mathcal{V} = \{v_1, v_2, \ldots, v_T\}$, where $T$ denotes the total number of visits and each visit $v_t$ contains a subset of medical codes, $v_t = \{c_{t,1}, \ldots, c_{t,N_{v_t}}\}, c_{t,i} \in \mathcal{C}$, where $N_{v_t}$ denotes the number of codes in $v_t$. EHR-based predictive analysis aims to predict future health outcomes given a patient's clinical trajectory $\mathcal{V}$. Typical tasks include predicting hospital readmission risk or estimating the set of diagnoses at the subsequent hospital visit. See Appendix B for a complete table of notations.

## 3 METHODS

In this section, we first introduce the overall architecture of our proposed model, DT-BEHRT, which consists of four main components: the Sequence Representation (SR) module, the Disease Aggregation (DA) module, the Disease Progression (DP) module, and the Patient Representation (PR) module (see Figure 1). Each module is designed to capture complementary aspects of a patient's evolving health trajectory, ranging from fine-grained event encoding to organ/system-level abstraction, temporal progression, and global patient summarization. We then present a novel pre-training framework specifically tailored to this architecture, including Global Code Masking Prediction (GCMP)

and Ancestor Code Prediction (ACP), which facilitates alignment across modules and improves the quality of patient representations for downstream predictive tasks.

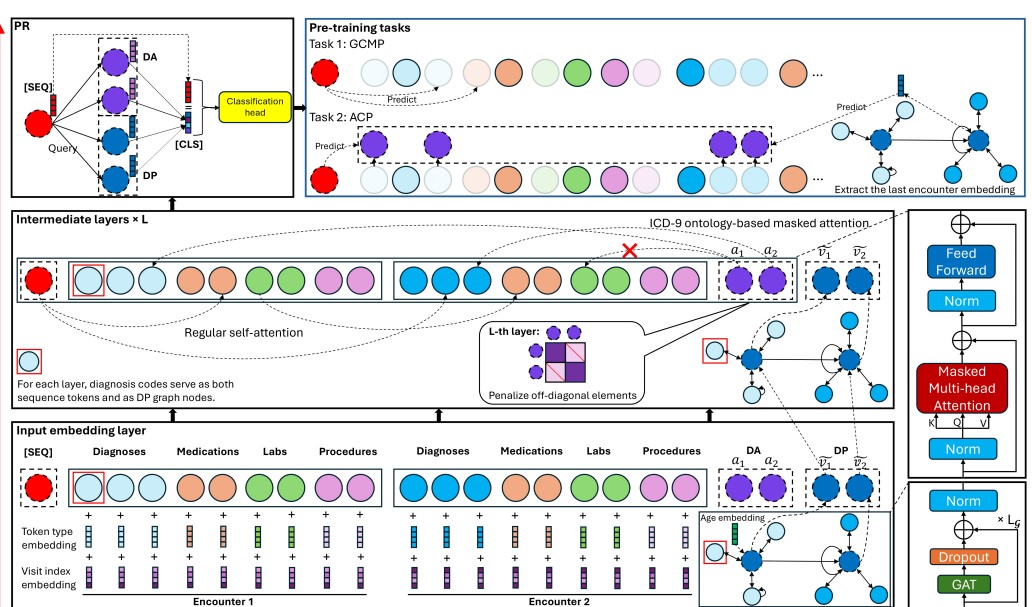

Figure 1: The architecture of DT-BEHRT. Each layer includes a Sequence Representation (SR), Disease Aggregation (DA), and Disease Progression (DP) module. The Patient Representation (PR) is derived via sequence-guided attention. The pre-training framework includes Global Code Masking Prediction (GCMP) and Ancestor Code Prediction (ACP).

### 3.1 SEQUENCE REPRESENTATION

The input to our model is a patient's medical code sequence $\mathcal{V}$, which is composed of $T$ hospital visits. Each token $c$ corresponds to a medical code drawn from the vocabulary $\mathcal{C}$, as defined in Section 2. To enrich the token representation, we incorporate two additional embeddings, similar as BEHRT (Li et al., 2020): a code-type embedding, $\boldsymbol{e}_{type(c)}$, which specifies whether a token belongs to diagnosis, medication, laboratory test, or procedure categories, and a visit-index embedding, $\boldsymbol{e}_{visit(c)}$, which encodes the relative temporal position of each visit in the patient's trajectory. The final token representation is obtained by summation: $\boldsymbol{h}_c^{(0)} = \boldsymbol{e}_c + \boldsymbol{e}_{type(c)} + \boldsymbol{e}_{visit(c)}$. Within a single visit, we make no assumptions about the ordering of codes, since the recorded timestamps of events within a single visit may not reflect their true temporal order. Following the BERT-style architecture, we prepend a special token [SEQ] to $\mathcal{V}$, which is designed to summarize the entire sequence. The input sequence is then processed by a stack of $L$ pre-normalization Transformer layers:

$$\boldsymbol{H}^{(0)} = \left[ \boldsymbol{h}_{[\text{SEQ}]}^{(0)} \,\|\, \boldsymbol{h}_{c_{1,1}}^{(0)} \,\|\, \dots \,\|\, \boldsymbol{h}_{c_{1,N_{v_1}}}^{(0)} \dots \,\|\, \boldsymbol{h}_{c_{T,1}}^{(0)} \dots \,\|\, \boldsymbol{h}_{c_{T,N_{v_T}}}^{(0)} \right], \tag{1}$$

$$\widetilde{\boldsymbol{H}}^{(l)} = \boldsymbol{H}^{(l)} + \text{MMHSA}\left( \text{LN}\left( \boldsymbol{H}^{(l-1)} \right), \boldsymbol{M} \right), \tag{2}$$

$$\boldsymbol{H}^{(l)} = \widetilde{\boldsymbol{H}}^{(l)} + \text{FFN}\left( \text{LN}\left( \widetilde{\boldsymbol{H}}^{(l)} \right) \right), \ \ 1 \le l \le L, \tag{3}$$

where $\|$ is the vector concatenation operator, $\text{MMHSA}(\cdot, \cdot)$ denotes the masked multi-head self-attention with attention mask $\boldsymbol{M}$ (to be introduced in the next subsection), $\text{FFN}(\cdot)$ is the position-wise feed-forward network, and $\text{LN}(\cdot)$ is the layer normalization. Thus, $\boldsymbol{H}^{(l)} \in \mathbb{R}^{(1+N_V) \times \text{d}}$, $N_V = \sum_{t=1}^{T} N_{v_t}$, is the hidden representation of all tokens at layer $l$, where $\text{d}$ is the hidden size.

### 3.2 DISEASE AGGREGATION

The ICD-9 ontology organizes diagnosis codes into nineteen top-level ancestor codes, or "chapters", denoted as $(\mathcal{J} = \{1, \dots, 19\})$, each corresponding to a specific organ/system-level dis-

ease class (e.g., cardiovascular diseases, see Appendix C for details) (CDC, 2013). Leveraging this rich hierarchical structure, we introduce a set of DA tokens, $\mathcal{A} = \{a_j : j \in \mathcal{J}\}$, with one token per top-level chapter, to summarize the progression and interactions of diseases within the same organ/system across visits, enabling the model to capture higher-level semantic patterns that extend beyond individual diagnosis codes. Let $\mathrm{Anc} : \mathcal{D} \rightarrow \mathcal{J}$ map each diagnosis code to its unique ICD-9 chapter. Define, for each $j \in \mathcal{J}, \mathcal{D}_j = \{d \in \mathcal{D} : \mathrm{Anc}(d) = j\} \subseteq \mathcal{D}$, and $\mathrm{anc}_{\mathcal{V}}(j) := |\mathrm{Supp}(\mathcal{V}) \cap \mathcal{D}_j|$ as the number of distinct codes from $\mathcal{D}_j$ that appear in the patient trajectory $\mathcal{V}$. Whenever $\mathrm{anc}_{\mathcal{V}}(j) \geq k$, where $k$ is a threshold hyperparameter, we append the DA token $a_j$ to the end of the visit-major vector $\boldsymbol{V} = \left( [\mathrm{SEQ}], c_{1,1}, \ldots, c_{1,N_{v_1}}, \ldots, c_{T,1}, \ldots, c_{T,N_{v_T}} \right)$, flattened from the trajectory $\mathcal{V}$ in visit order. The resulting concatenated vector is as follows: $\boldsymbol{V_a} = [\boldsymbol{V} \,\|\, \boldsymbol{a}_{\mathcal{V}}]$, where $\boldsymbol{a}_{\mathcal{V}} = \left( a_{j_1}, \ldots, a_{j_{N_a}} \right), j_1 < \ldots < j_{N_a}$ are the categories satisfying the threshold condition, and $N_a = |\boldsymbol{a}_{\mathcal{V}}|$. For the concatenated vector $\boldsymbol{V_a}$, we apply an attention mask, $\boldsymbol{M} \in \mathbb{R}^{(1+N_V+N_a) \times (1+N_V+N_a)}$, that restricts attention to diagnosis codes within each DA token's ICD-9 chapter and to the token itself:

$$
\boldsymbol{M}[l, m] = \begin{cases} 0, & \text{if } l = m \text{ and } l > N_V + 1, \\ 0, & \text{if } l \leq N_V + 1 \text{ and } m \leq N_V + 1, \\ 0, & \text{if } l > N_V + 1, \ m \leq N_V + 1, \text{ and } \boldsymbol{V_a}[m] \in \mathcal{D}_{\phi(l)}, \\ -\infty, & \text{otherwise.} \end{cases} \tag{4}
$$

where $\phi(l)$ denote the ICD-9-chapter index of the DA token placed at row $l$ (for $l > N_V + 1$). Equivalently, $a_{\phi(l)} = a_{j_{(l-N_V-1)}}$. A sample attention mask can be found in Appendix D.

In the SR module, the attention among diagnosis codes is unconstrained, which may lead to redundancy when similar information is aggregated through DA tokens. To encourage the DA tokens to encode rich and diverse information, we introduce a token-level covariance regularization. Formally, we extract $\boldsymbol{Z} \in \mathbb{R}^{N_a \times \mathrm{d}}$ from $\boldsymbol{H}^{(L)}$. The regularization term $\ell_{cov}$ is defined as follows:

$$
\ell_{cov} = \frac{1}{(N_a - 1)^2} \sum_{i \neq j}^{N_a} \left( \mathrm{Cov}(\boldsymbol{Z})[i, j] \right)^2, \tag{5}
$$

where $\mathrm{Cov}(\boldsymbol{Z}) = \frac{1}{\mathrm{d} - 1} \sum_{j=1}^{\mathrm{d}} \left( \boldsymbol{Z}_{:,j} - \bar{\boldsymbol{Z}}_j \right) \left( \boldsymbol{Z}_{:,j} - \bar{\boldsymbol{Z}}_j \right)^T$, and $\bar{\boldsymbol{Z}}_j = \frac{1}{\mathrm{d}} \sum_{j=1}^{\mathrm{d}} \boldsymbol{Z}_{:,j}$. This regularization term encourages the off-diagonal elements of the covariance matrix $\mathrm{Cov}(\boldsymbol{Z})$ to approach zero, thereby compelling the DA tokens to capture decorrelated organ/system-level abstractions.

## 3.3 DISEASE PROGRESSION

We construct a heterogeneous graph $\mathcal{G} = (\mathcal{U}, \mathcal{E}, \mathcal{X})$ to model a patient's disease progression and better capture potential development trends. Here, $\mathcal{U}, \mathcal{E}, \mathcal{X}$ denote the node set, the edge set, and the node feature set, respectively. The graph consists of $T$ virtual visit nodes, each corresponding to one hospital visit, together with the diagnosis nodes associated with that visit. Formally, $\mathcal{U} = \{\tilde{v}_1, \ldots, \tilde{v}_T\} \cup \left\{ \tilde{d}_{i,t} \mid i = 1, \ldots, N_{d_t}, t = 1, \ldots, T \right\}$, where $N_{d_t}$ denotes the number of diagnosis codes for visit $t$. We hereafter refer to the virtual visit nodes $\tilde{v}_t$ as DP nodes, emphasizing their role in encoding disease development trends through graph learning. Directed edges are added from each DP node to its diagnosis nodes, $\tilde{d}_{t,i}$, while DP nodes are connected sequentially in temporal order through forward-directed edges. In addition, self-loops are introduced for DP nodes starting from the second DP node, ensuring that these nodes can preserve their own information during message passing. Formally, $\mathcal{E} = \left\{ (\tilde{v}_t \leftrightarrow \tilde{d}_{t,i}) \mid i = 1, \ldots, N_{d_t}, t = 1, \ldots, T \right\} \cup \left\{ (\tilde{d}_{t,i} \rightarrow \tilde{d}_{t,i}) \mid i = 1, \ldots, N_{d_t}, t = 1, \ldots, T \right\} \cup \left\{ (\tilde{v}_t \rightarrow \tilde{v}_{t+1}) \mid t = 1, \ldots, T-1 \right\} \cup \left\{ (\tilde{v}_t \rightarrow \tilde{v}_t) \mid t = 2, \ldots, T \right\}$. For layer $l = 1$, DP visit node features are initialized with patient age embeddings, $\boldsymbol{e}_{Age(t)}$, while disease node features come from the embedding of the corresponding diagnosis code in the SR module, $\boldsymbol{h}_{d_{t,i}}^{(0)}$. In higher layers $(2 \leq l \leq L)$, node features are updated through message passing: visit node features are taken from

the previous DP layer, and diagnosis node features from the previous SR layer.

$$\mathcal{X}^{(l)} = \left\{ \boldsymbol{h}_{\tilde{d}_{t,i}}^{(l-1)}, \boldsymbol{h}_{\tilde{v}_t}^{(l-1)} \mid i = 1, \ldots, N_{d_t}, t = 1, \ldots, T \right\},$$

where $\boldsymbol{h}_{\tilde{d}_{t,i}}^{(l)} = \boldsymbol{h}_{d_{t,i}}^{(l)}$ and $\boldsymbol{h}_{\tilde{v}_t}^{(0)} = \boldsymbol{e}_{Age(t)}$. Furthermore, the representations of DP nodes, $\boldsymbol{h}_{\tilde{v}_t}^{(l)}$, is updated by a graph attention network (GAT) layer (Veličković et al., 2017) as follows:

$$Message_{\tilde{d}_{t,i} \to \tilde{v}_t}^{(l)} = \sum_{i=1}^{N_{d_t}} \text{GAT}^{(l)} \left( \tilde{d}_{t,i} \to \tilde{v}_t \right), \tag{6}$$

$$Message_{\{\tilde{v}_{t-1}, \tilde{v}_t\} \to \tilde{v}_t}^{(l)} = \sum_{v' \in \{\tilde{v}_{t-1}, \tilde{v}_t\}} \text{GAT}^{(l)} \left( v' \to \tilde{v}_t \right), \tag{7}$$

$$\tilde{\boldsymbol{h}}_{\tilde{v}_t}^{(l)} = Message_{\tilde{d}_{t,i} \to \tilde{v}_t}^{(l)} + Message_{\{\tilde{v}_{t-1}, \tilde{v}_t\} \to \tilde{v}_t}^{(l)}, \text{ and } \boldsymbol{h}_{\tilde{v}_t}^{(l)} = \text{LN} \left( \tilde{\boldsymbol{h}}_{\tilde{v}_t}^{(l)} + \boldsymbol{h}_{\tilde{v}_t}^{(l-1)} \right). \tag{8}$$

We then stack $L_{\mathcal{G}}$ such GAT blocks within each layer's DP module, allowing each DP node representation $\boldsymbol{h}_{\tilde{v}_t}^{(l)}$ to incorporate information from visits up to $L_{\mathcal{G}}$-hops away (e.g., from $\boldsymbol{h}_{\tilde{v}_{(t-L_{\mathcal{G}})}}^{(l)}$).

### 3.4 PATIENT REPRESENTATION

At the final layer $L$, we integrate three complementary sources of information. The representation of the [SEQ] token, $\boldsymbol{h}_{[\text{SEQ}]}^{(L)}$, summarizes the entire medical code sequence $\mathcal{V}$ of a patient. The representations of the DA tokens, $\left\{ \boldsymbol{h}_{a_j}^{(L)} \mid j \in \mathcal{J}, \text{anc}_{\mathcal{V}}(j) \geq k \right\}$, capture the progression and interactions of diseases within the same organ/system across visits. The representations of the DP tokens, $\left\{ \boldsymbol{h}_{\tilde{v}_t}^{(L)} \mid t = 1, \ldots, T \right\}$, updated through GAT blocks, model potential disease development trends along the temporal trajectory. By integrating these components, we derive the final patient representation vector, $\boldsymbol{h}_{[\text{CLS}]}$, which serves as the input for downstream predictive tasks. We design an attention-based mechanism that leverages sequence-level information to differentiate the relative importance of DA tokens and DP tokens. We derive $\boldsymbol{h}_{[\text{CLS}]}$ by:

$$\boldsymbol{h}_{[\text{CLS}]} = \left[ \boldsymbol{h}_{[\text{SEQ}]}^{(L)} \mid\mid \text{Attn} \left( \boldsymbol{h}_{[\text{SEQ}]}^{(L)}, \left\{ \boldsymbol{h}_{a_j}^{(L)} \mid j \in \mathcal{J}, \text{anc}_{\mathcal{V}}(j) \geq k \right\} \cup \left\{ \boldsymbol{h}_{\tilde{v}_t}^{(L)} \mid t = 1, \ldots, T \right\} \right) \right], \tag{9}$$

where $\text{Attn}(\cdot, \cdot)$ denotes the attention pooling mechanism.

### 3.5 PRE-TRAINING FRAMEWORK

To fully exploit the information contained in the dataset and to enhance alignment across the SR, DA, and DP modules, we design a novel pre-training framework tailored to our model architecture.

*A. Global Code Masking Prediction:* Inspired by Med-BERT (Rasmy et al., 2021) and HEART (Huang et al., 2024), we adopt masked token prediction as one of the pre-training tasks. However, our design differs in key aspects. Since the timestamp order within a visit may not reflect true occurrences and repeated codes across visits may create shortcuts, we instead encourage the model to capture co-occurrence semantics at the trajectory level, which better encodes patterns such as comorbidities and treatment pathways. Specifically, given a patient's medical code sequence, we first identify all unique codes. For each code type (i.e., diagnosis, medication, laboratory test, and procedure), we independently sample codes for masking at the unique-code level with rate $\alpha$. Once a code is selected, all of its occurrences in $\mathcal{V}$ are masked. Then, $\boldsymbol{h}_{[\text{CLS}]}$ is required to predict the masked codes across all four categories simultaneously, encouraging the learned representation to be broadly generalizable to diverse downstream tasks. The loss term $\ell_{\text{mask}}$ is defined as follows:

$$\ell_{\text{mask}} = \frac{1}{4} \sum_{\tau \in \mathcal{T}} \text{BCE} \left( P_\tau, Y_{\text{mask},\tau} \right), \tag{10}$$

where $P_\tau = \sigma \left( \text{Linear}_\tau \left( \boldsymbol{h}_{[\text{CLS}]} \right) \right)$ denotes the prediction heads for code type $\tau$, with $\tau \in \mathcal{T} = \{ \tau_{\mathcal{D}}, \tau_{\mathcal{M}}, \tau_{\mathcal{L}}, \tau_{\mathcal{P}} \}$, corresponding to the four code types. The operator $\text{Linear}_\tau(\cdot)$ is the linear layer associated with the prediction head $\tau$, $\sigma(\cdot)$ denotes sigmoid activation, $\text{BCE}(\cdot, \cdot)$ is the binary cross entropy loss function, and $Y_{\text{mask},\tau}$ is the masked token label of code type $\tau \in \mathcal{T}$.

*B. Ancestor Code Prediction:* In our architecture, the DA module explicitly incorporates ICD-9 high-level chapters, while the SR and DP modules are not directly exposed to this ontology information. This asymmetry may lead to misalignment when constructing the final patient representation $\boldsymbol{h}_{[\text{CLS}]}$, where $\boldsymbol{h}_{[\text{SEQ}]}^{(L)}$ serves as the query in the attention mechanism, potentially hurting downstream performance. To address this issue and make the other two modules aware of the ontology structure, we introduce an auxiliary ancestor code prediction task. Specifically, for each masked diagnosis code in the masked token prediction task, we require the model to predict its ancestor code in the ICD-9 ontology. The predictions are made from two perspectives: a) using the $\boldsymbol{h}_{[\text{SEQ}]}^{(L)}$ from the SR module, and b) using the representation of the last DP token, $\boldsymbol{h}_{\tilde{v}_T}^{(L)}$, which partially serves as a summary of the DP graph. This design encourages the representations across modules to jointly understand ontology-level knowledge, thereby promoting better alignment. The loss term $\ell_{\text{anc}}$ is defined as $\ell_{\text{anc}} = \ell_{\text{anc,SR}} + \ell_{\text{anc,DP}}$, where

$$\ell_{\text{anc,SR}} = \text{BCE}\left(\sigma\left(\text{Linear}\left(\boldsymbol{h}_{[\text{SEQ}]}^{(L)}\right)\right), \text{Anc}\left(Y_{\text{mask},\tau_{\mathcal{D}}}\right)\right), \tag{11}$$

$$\ell_{\text{anc,DP}} = \text{BCE}\left(\sigma\left(\text{Linear}\left(\boldsymbol{h}_{\tilde{v}_T}^{(L)}\right)\right), \text{Anc}\left(Y_{\text{mask},\tau_{\mathcal{D}}}\right)\right). \tag{12}$$

### 3.6 LEARNING OBJECTIVES

During the pre-training phase, the model is optimized with a joint objective that combines masked token prediction, ancestor node prediction, and DA token decorrelation. The strengths of the ancestor node prediction and DA decorrelation penalties are controlled by $\lambda_{\text{anc}}$ and $\lambda_{\text{cov}}$, respectively. Formally, $\ell_{\text{pt}} = \ell_{\text{mask}} + \lambda_{\text{anc}}\ell_{\text{anc}} + \lambda_{\text{cov}}\ell_{\text{cov}}$. During the fine-tuning phase, the learning objective is given by $\ell_{\text{ft}} = \ell_{\text{task}} + \lambda_{\text{cov}}\ell_{\text{cov}}$, where $\ell_{\text{task}} = \text{BCE}\left(\sigma\left(\text{Linear}\left(\boldsymbol{h}_{[\text{CLS}]}\right)\right), Y_{\text{task}}\right)$ for ground truth label, $Y_{\text{task}}$, of the downstream task. The detailed pseudocode for DT-BEHRT pre-training and fine-tuning is provided in Algorithm 1 in Appendix G.

## 4 EXPERIMENTS

### 4.1 DATASETS

We conduct experiments on the MIMIC-III (Johnson et al., 2016), MIMIC-IV (Johnson et al., 2023) and eICU (Pollard et al., 2018) datasets, three publicly available EHR databases hosted on PhysioNet (https://physionet.org/). The data preprocessing steps follow HEART (Huang et al., 2024), with details provided in the Appendix E. To comprehensively evaluate our model, we examine three standard outcome prediction tasks on the MIMIC-III and MIMIC-IV cohorts: in-hospital mortality, prolonged length of stay (PLOS; defined as hospitalization exceeding 7 days), and readmission. In addition, we perform phenotyping prediction for a set of acute, chronic, and mixed conditions at the next hospital encounter within 12 months, formulated as a multi-label classification task and aligned with the experimental setup in DrFuse (Yao et al., 2024). For the eICU dataset, we only evaluate ICU mortality and PLOS.

### 4.2 BASELINES

We comprehensively compare our model with state-of-the-art EHR-based predictive models across three categories: graph-based models, sequence-based models, and graph-enhanced sequence models. The implementation details of our model can be found in Appendix F. Since our method falls into the category of graph-enhanced sequence models, we place particular emphasis on recent advances in sequence-based and graph-enhanced sequence approaches. For graph-based approach, we select HypEHR (Xu et al., 2023) as a representative baseline, as it reflects the most recent endeavor in using hypergraph to capture the high-order interaction between codes and visits. For sequence-based approach, BEHRT (Li et al., 2020) represents one of the earliest transformer-based models for EHR. It organizes a patient's historical diagnoses into a sentence fed into a transformer. Med-BERT (Rasmy et al., 2021) extends the BERT framework to pre-train on large-scale EHR data. ExBEHRT (Rupp et al., 2023) extends BEHRT (Li et al., 2020) by integrating additional types of codes through vertical summation of their embeddings. For graph-enhanced sequence approach, G-BERT (Shang et al., 2019) embeds hierarchical information of diagnosis and medication codes with a GAT and

encodes their sequences using BERT (Devlin et al., 2019). HEART (Huang et al., 2024) enriches medical code representations with heterogeneous relation embeddings that explicitly parameterize pairwise correlations between entities, and further enhances hospital visit representations by connecting them as a graph and applying a modified GAT. All baselines are trained and evaluated under exactly the same cohort construction, inclusion criteria, and prediction windows.

## 4.3 EXPERIMENT RESULTS

### 4.3.1 PERFORMANCE ON GENERAL OUTCOME PREDICTION

Table 1 shows that DT-BEHRT generally outperforms all baselines across tasks on both datasets. The largest performance gain is observed on the readmission task, which is known to be particularly challenging in EHR-based prediction due to the heterogeneous and multifactorial causes of readmission. On the smaller dataset MIMIC-III, our model shows a clear advantage, while on the larger dataset MIMIC-IV, this advantage becomes less pronounced, suggesting that larger data availability partially compensates for the modeling gaps of baseline methods. On the eICU dataset, DT-BEHRT also attains the best overall performance, and the consistent results across MIMIC and eICU indicate that the model generalizes reasonably well across different clinical databases. The hypergraph-based approach HypEHR (Xu et al., 2023) exhibits high instability on the readmission task in MIMIC-III. A possible reason is that, with limited data and a large vocabulary, the resulting hypergraph suffers from low hyperedge density, weakening its ability to capture reliable high-order interactions as well as temporal dependencies—particularly for readmission, which requires robust modeling of long-term disease progression (Pham et al., 2016). Although HEART (Huang et al., 2024) employs a hierarchical design from codes to visits, it still underperforms compared to DT-BEHRT on the readmission task. One reason may be that it does not explicitly differentiate the importance of diagnosis codes from other code types, leading to incomplete transmission of critical progression information during the transition from visit-level to patient-level representations.

Table 1: Results of general outcome prediction tasks.

| Models | | | G-BERT | BEHRT | Med-BERT | HypEHR | ExBEHRT | HEART | DT-BEHRT |
|---|---|---|---|---|---|---|---|---|---|
| **MIMIC-III** | Mortality | F1 | 59.24±0.46 | 68.60±0.43 | 67.91±1.08 | 70.04±0.70 | 73.66±1.09 | 74.77±1.26 | **76.03±0.28** |
| | | AUROC | 86.25±0.82 | 87.23±0.27 | 87.94±0.53 | 88.55±0.39 | 90.72±0.30 | **92.13±0.36** | 92.09±0.15 |
| | | AUPRC | 72.13±1.51 | 75.33±0.33 | 75.28±1.01 | 76.39±1.06 | 81.44±0.62 | 82.76±0.63 | **84.50±0.19** |
| | PLOS | F1 | 69.62±1.42 | 70.38±0.74 | 72.02±1.43 | 72.73±0.09 | 74.73±0.70 | 75.44±1.47 | **76.37±0.49** |
| | | AUROC | 72.96±1.20 | 73.71±0.98 | 77.25±0.53 | 78.89±0.33 | 82.11±0.71 | 82.99±0.40 | **84.13±0.26** |
| | | AUPRC | 72.48±1.27 | 72.49±1.47 | 76.98±0.56 | 78.70±0.28 | 83.52±0.79 | 83.83±0.59 | **85.00±0.22** |
| | Readmission | F1 | 60.84±1.01 | 53.64±1.56 | 66.52±0.92 | 48.09±3.57 | 63.08±0.82 | 68.77±0.36 | **70.59±0.34** |
| | | AUROC | 67.40±0.65 | 64.79±0.44 | 76.66±0.57 | 68.28±0.30 | 73.68±0.70 | 77.68±0.79 | **80.30±0.14** |
| | | AUPRC | 57.19±0.77 | 51.59±0.47 | 62.90±1.50 | 56.00±0.11 | 62.13±0.91 | 64.05±1.46 | **69.62±0.20** |
| **MIMIC-IV** | Mortality | F1 | 58.06±1.01 | 67.22±1.06 | 66.55±2.38 | 65.27±2.30 | 70.25±0.72 | 70.52±0.86 | **70.89±0.53** |
| | | AUROC | 93.15±0.62 | 94.84±0.20 | 94.98±0.40 | 95.27±0.18 | 96.19±0.13 | 96.12±0.12 | **96.21±0.12** |
| | | AUPRC | 68.52±3.78 | 71.66±0.93 | 71.52±2.53 | 71.63±0.78 | 77.00±0.86 | 76.94±0.59 | **78.35±0.37** |
| | PLOS | F1 | 61.27±1.02 | 61.47±0.53 | 63.38±1.03 | 61.77±0.75 | 67.64±0.42 | 67.07±1.27 | **68.04±0.54** |
| | | AUROC | 77.34±0.66 | 78.62±0.52 | 81.89±0.29 | 81.00±0.13 | **84.99±0.21** | 84.63±0.35 | 84.98±0.09 |
| | | AUPRC | 66.68±0.84 | 66.19±0.97 | 70.82±0.37 | 69.39±0.24 | **75.97±0.54** | 74.48±0.96 | 74.78±0.23 |
| | Readmission | F1 | 82.13±0.35 | 82.76±0.43 | 83.19±0.55 | 82.80±0.18 | 83.21±0.61 | 83.68±0.35 | **84.18±0.08** |
| | | AUROC | 65.38±0.34 | 62.32±0.23 | 68.51±0.66 | 66.07±0.27 | 68.41±0.38 | 68.93±1.11 | **72.08±0.25** |
| | | AUPRC | 71.49±0.31 | 78.23±0.17 | 81.89±0.46 | 80.50±0.16 | 81.86±0.16 | 82.07±0.53 | **84.85±0.14** |
| **eICU** | Mortality | F1 | 66.46±0.73 | 60.01±1.06 | 75.04±1.73 | 75.83±0.78 | 71.21±0.14 | 73.08±0.34 | **81.27±0.21** |
| | | AUROC | 89.28±0.72 | 78.11±0.22 | 91.04±0.47 | 90.39±0.48 | 87.53±0.24 | 88.65±0.12 | **93.73±0.06** |
| | | AUPRC | 77.48±2.65 | 64.04±0.45 | 80.56±1.77 | 83.87±0.82 | 78.36±0.50 | 79.95±0.14 | **88.58±0.13** |
| | PLOS | F1 | 65.73±1.22 | 49.04±1.04 | 67.98±1.23 | 67.25±1.19 | 60.77±2.90 | 69.71±0.64 | **72.49±0.23** |
| | | AUROC | 76.44±0.93 | 63.12±0.58 | 80.86±0.41 | 82.04±1.09 | 77.77±0.84 | 82.93±0.28 | **85.84±0.11** |
| | | AUPRC | 70.76±1.05 | 50.58±0.98 | 75.08±0.47 | 75.61±1.54 | 68.83±1.44 | 77.53±0.33 | **81.07±0.22** |

To further assess the robustness of DT-BEHRT, we evaluate its performance across clinically relevant patient subgroups on the MIMIC-III dataset (Figure 2). The analysis includes nine conditions: hypertension, diabetes mellitus, chronic kidney disease (CKD), coronary artery disease (CAD), heart failure, chronic obstructive pulmonary disease (COPD), liver disease, and cancer. Across these subgroups, DT-BEHRT generally achieves the competitive performance on mortality and PLOS, while for readmission it attains the best performance across all subgroups.

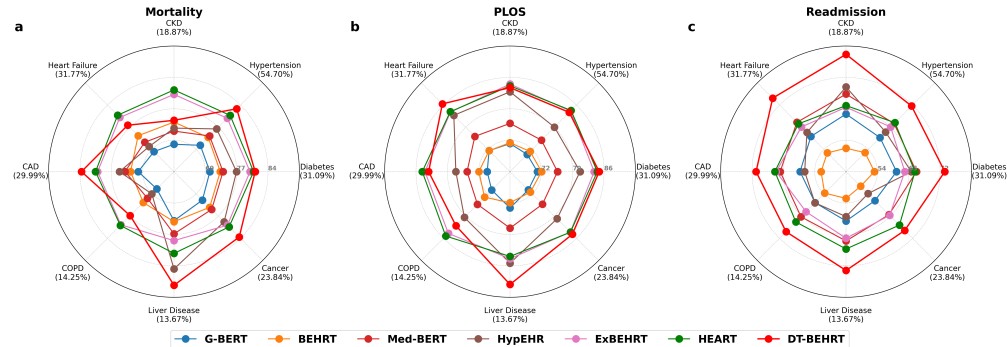

Figure 2: Subgroup performance radar plots for **a** mortality, **b** PLOS, and **c** readmission prediction tasks across major comorbidity groups.

### 4.3.2 PERFORMANCE ON PHENOTYPING PREDICTION

For phenotyping prediction, we evaluate the top three models from the general outcome prediction task—ExBEHRT (Rupp et al., 2023), HEART (Huang et al., 2024), and DT-BEHRT. Using macro-AUPRC as the overall metric for multi-phenotype prediction, DT-BEHRT consistently achieves the best performance in both the full cohort and in patients with three or more hospital visits (Table 2). The performance gain is particularly pronounced in the latter, suggesting that DT-BEHRT effectively captures disease progression in patients with strong temporal dependencies characterized by repeated hospitalizations.

Table 2: Results of phenotyping prediction tasks.

| | Prevalence | All patients | | | Patients with ≥ 3 visits | | |
|---|---|---|---|---|---|---|---|
| | | ExBEHRT | HEART | DT-BEHRT | ExBEHRT | HEART | DT-BEHRT |
| **MIMIC-III** | | | | | | | |
| Acute and unspecified renal failure | 16.00% | **49.96±0.96** | 47.29±1.81 | 44.62±1.91 | 46.98±4.75 | 46.52±2.52 | **54.26±4.55** |
| Acute cerebrovascular disease | 0.90% | 3.95±2.30 | 3.21±0.91 | **7.53±2.11** | 2.11±0.96 | 3.82±2.99 | **17.74±33.72** |
| Acute myocardial infarction | 3.70% | **18.49±1.40** | 16.61±1.41 | 17.27±1.18 | **17.81±7.67** | 16.21±4.25 | 11.28±3.64 |
| Cardiac dysrhythmias | 20.10% | 74.68±1.48 | **74.98±1.47** | 72.81±1.87 | 71.35±6.60 | **77.02±6.09** | 70.47±2.39 |
| Chronic kidney disease | 12.40% | **78.88±0.96** | 76.96±2.74 | 77.21±2.11 | 75.09±5.75 | 81.45±6.62 | **88.72±1.24** |
| Chronic obstructive pulmonary disease | 6.40% | 42.85±1.72 | 43.31±3.21 | **43.66±2.79** | 45.34±6.44 | 40.24±8.77 | 44.68±6.92 |
| Conduction disorders | 1.40% | 4.19±0.62 | 4.24±0.87 | **4.54±1.63** | 13.35±13.67 | 3.32±1.53 | 8.00±5.39 |
| Congestive heart failure; non-hypertensive | 20.10% | **75.09±1.93** | 74.27±1.04 | 72.06±1.54 | 75.55±4.33 | 74.41±4.27 | **80.96±2.23** |
| Coronary atherosclerosis and related | 12.10% | 61.11±1.70 | **61.87±1.33** | 60.20±1.83 | 61.94±4.15 | 61.88±3.66 | **66.90±4.77** |
| Disorders of lipid metabolism | 13.70% | **59.13±1.37** | 57.82±1.83 | 56.56±2.80 | 55.65±8.81 | **62.16±10.29** | 55.09±3.78 |
| Essential hypertension | 18.90% | **67.85±1.73** | 64.75±2.07 | 63.00±2.23 | 67.50±4.39 | **68.06±5.32** | 57.55±4.57 |
| Fluid and electrolyte disorders | 21.00% | 48.32±1.37 | 46.37±0.59 | **48.45±1.50** | 44.38±3.28 | 46.60±3.22 | **57.55±2.99** |
| Gastrointestinal hemorrhage | 3.60% | 8.75±0.52 | 9.02±0.59 | **12.06±1.95** | 12.28±6.66 | 10.37±6.11 | **17.18±4.97** |
| Hypertension with complications | 11.50% | 72.17±2.10 | **72.26±3.77** | 71.16±3.22 | 66.82±9.06 | 78.94±1.88 | **82.88±2.60** |
| Other liver diseases | 0.90% | 4.76±1.46 | **5.71±3.23** | 3.11±0.76 | **11.13±18.13** | 3.55±3.08 | 4.14±2.49 |
| Other lower respiratory disease | 21.40% | 56.36±0.80 | 56.50±1.10 | **57.96±1.95** | 58.56±4.02 | 58.45±4.73 | **71.85±3.15** |
| Pneumonia | 7.10% | 16.17±1.79 | 16.51±1.16 | **17.57±1.45** | 15.60±2.00 | **20.53±4.65** | 16.78±3.73 |
| Septicemia (except in labor) | 11.70% | 35.34±0.85 | 33.50±1.19 | **36.25±1.51** | 32.63±4.14 | 31.36±5.10 | **43.89±3.59** |
| **Macro AUPRC** | / | 43.22±0.69 | 42.51±0.84 | **43.45±0.46** | 43.38±1.81 | 43.61±0.57 | **48.15±2.45** |
| **MIMIC-IV** | | | | | | | |
| Acute and unspecified renal failure | 12.50% | **42.79±1.51** | 42.14±1.92 | 41.98±1.79 | 42.45±3.02 | 39.03±2.57 | **48.81±2.06** |
| Acute cerebrovascular disease | 0.40% | 2.08±0.88 | 1.12±0.15 | **2.78±0.62** | **7.21±9.28** | 1.31±0.56 | 0.32±0.17 |
| Acute myocardial infarction | 2.10% | **15.68±1.55** | 14.02±2.20 | 12.58±0.81 | **21.77±4.61** | 12.65±3.72 | 10.91±2.44 |
| Cardiac dysrhythmias | 14.30% | 71.91±0.92 | 72.53±0.99 | **73.17±1.45** | 70.13±2.93 | 72.64±3.43 | **76.01±1.21** |
| Chronic kidney disease | 13.70% | 85.56±1.17 | 85.72±2.35 | **86.15±1.56** | 85.51±3.15 | 84.69±2.19 | **89.53±0.85** |
| Chronic obstructive pulmonary disease | 4.60% | 49.82±1.52 | 52.10±1.21 | **52.23±1.68** | **55.95±3.20** | 51.11±4.71 | 50.04±2.64 |
| Conduction disorders | 1.20% | 6.45±0.96 | **8.53±2.21** | 7.64±0.73 | 8.59±4.33 | **12.46±13.86** | 5.13±1.86 |
| Congestive heart failure; non-hypertensive | 12.50% | 75.76±1.83 | **77.93±1.19** | 77.53±1.23 | 74.39±2.29 | 76.86±1.24 | **87.03±1.64** |
| Coronary atherosclerosis and related | 13.50% | 80.61±0.45 | 80.24±0.42 | **81.87±1.18** | 80.64±1.18 | 77.45±2.25 | **81.84±1.48** |
| Disorders of lipid metabolism | 19.80% | 75.40±0.71 | 75.69±0.92 | **76.29±0.87** | 76.34±1.58 | 75.72±2.44 | **80.08±1.52** |
| Essential hypertension | 21.70% | 76.19±0.72 | 76.91±1.84 | **79.20±1.11** | 75.96±2.50 | 77.91±2.96 | **80.30±1.63** |
| Fluid and electrolyte disorders | 17.10% | **45.23±1.39** | 45.15±0.66 | 45.16±1.61 | 45.36±2.11 | 43.54±4.39 | **51.84±1.82** |
| Gastrointestinal hemorrhage | 2.20% | 5.92±0.28 | **7.09±1.31** | 6.66±0.66 | 6.32±1.87 | 8.65±3.75 | **9.09±1.85** |
| Hypertension with complications | 11.50% | 78.31±1.95 | 79.35±2.49 | **80.07±2.10** | 77.37±4.72 | 78.77±3.44 | **83.53±1.49** |
| Other liver diseases | 0.50% | 2.01±0.11 | **2.67±1.55** | 2.05±0.52 | 2.22±1.15 | **6.73±7.80** | 5.36±2.91 |
| Other lower respiratory disease | 9.20% | 34.60±1.41 | 35.05±1.33 | **35.20±2.17** | 34.84±2.56 | 36.79±4.02 | **46.80±2.18** |
| Pneumonia | 4.20% | **12.45±0.63** | 11.26±0.59 | 12.35±0.57 | 12.38±1.06 | 11.16±2.54 | **13.56±0.90** |
| Septicemia (except in labor) | 4.70% | **16.88±0.51** | 14.55±0.67 | 15.71±1.10 | 17.89±1.67 | 15.39±2.68 | **21.36±1.78** |
| **Macro AUPRC** | / | 43.20±0.41 | 43.45±0.65 | **44.57±0.08** | 44.18±0.26 | 43.49±0.93 | **47.23±0.28** |

### 4.3.3 ABLATION STUDY

As shown in Table 3, enabling the DA tokens without the covariance regularization loss (DA$^{w/o\ cov}$) yields only modest improvements. When the full DA module is enabled, the model demonstrates a clear benefit on the mortality prediction task, whereas the performance on the other two tasks remains largely unchanged or shows slight degradation. This observation is consistent with clinical intuition, as certain disease categories (e.g., cardiovascular diseases) are more directly associated with fatal outcomes compared to others (e.g., endocrine disorders). By summarizing diagnosis information through DA tokens and directly propagating them into the patient representation, the model is able to leverage this critical information more effectively. When enabling only the DP module, the results confirm our earlier findings on the MIMIC-III readmission task: modeling disease progression with forward-connected heterogeneous graphs provides the greatest benefit, as the DP module explicitly injects temporal dependencies into the final patient representation. Consistent with prior studies, the GCMP task–analogous to masked language modeling (MLM; Devlin et al. 2019)–serves as the primary source of performance gains. However, we also observe that a variant trained with GCMP alone, without the DA and DP modules, underperforms the full model, suggesting beneficial interactions between the architectural components and the pretraining objectives. In addition to the GCMP task, we introduce the novel ACP task, which yields the most pronounced improvements on the mortality prediction task.

Table 3: Ablation study for general outcome prediction tasks using MIMIC-III and MIMIC-IV.

| Variant | | | Performance | | | | | | | |
|---|---|---|---|---|---|---|---|---|---|---|
| | Architectures | DA$^{w/o\ cov}$ | × | ✓ | × | × | × | × | × | × |
| | | DA | × | × | ✓ | × | ✓ | × | ✓ | ✓ |
| | | DP | × | × | × | ✓ | ✓ | × | ✓ | ✓ |
| | Pre-training Tasks | GCMP | × | × | × | × | × | ✓ | ✓ | ✓ |
| | | ACP | × | × | × | × | × | × | × | ✓ |
| MIMIC-III | Mortality | F1 | 71.59±1.29 | 72.01±1.76 | 73.51±1.69 | 72.06±0.82 | 73.48±0.47 | 74.17±0.09 | 75.40±0.49 | **76.03±0.28** |
| | | AUROC | 89.18±1.34 | 90.11±0.98 | 90.34±1.00 | 89.55±0.20 | 90.44±0.38 | 91.41±0.32 | 91.72±0.28 | **92.09±0.15** |
| | | AUPRC | 80.04±1.69 | 79.82±1.97 | 81.55±1.61 | 80.35±0.41 | 81.65±0.57 | 82.21±0.34 | 83.70±0.71 | **84.50±0.19** |
| | PLOS | F1 | 75.07±0.19 | 75.27±0.89 | 74.37±1.24 | 75.34±0.51 | 75.52±0.40 | 76.17±0.11 | **77.19±0.27** | 76.37±0.49 |
| | | AUROC | 81.67±0.56 | 81.95±0.96 | 80.98±1.51 | 82.12±0.68 | 82.41±0.41 | 83.51±0.44 | **84.35±0.31** | 84.13±0.26 |
| | | AUPRC | 82.43±0.47 | 82.30±1.21 | 82.24±1.34 | 83.17±0.78 | 83.52±0.34 | 84.22±0.43 | **85.05±0.40** | 85.00±0.22 |
| | Readmission | F1 | 70.39±0.32 | 68.37±0.93 | 69.88±0.67 | 70.18±0.44 | 70.54±0.14 | 69.75±0.26 | 70.32±0.64 | **70.59±0.34** |
| | | AUROC | 79.42±0.36 | 78.78±0.34 | 79.30±0.44 | 79.98±0.18 | 79.99±0.16 | 79.90±0.22 | **80.49±0.18** | 80.30±0.14 |
| | | AUPRC | 67.77±1.21 | 68.96±0.58 | 68.32±0.59 | 69.16±0.59 | 69.22±0.17 | **69.94±0.43** | 69.84±0.29 | 69.62±0.20 |
| MIMIC-IV | Mortality | F1 | 63.84±2.09 | 65.70±1.13 | 66.37±0.73 | 65.89±2.30 | 67.25±0.84 | 67.78±0.70 | 68.58±0.33 | **70.89±0.53** |
| | | AUROC | 93.83±0.37 | 94.58±0.54 | 94.66±0.27 | 94.48±0.79 | 95.05±0.37 | 95.13±0.18 | 95.78±0.11 | **96.21±0.12** |
| | | AUPRC | 69.86±1.69 | 72.27±1.74 | 72.77±0.79 | 72.21±2.25 | 74.33±0.83 | 74.07±0.54 | 76.16±0.38 | **78.35±0.37** |
| | PLOS | F1 | 66.39±0.87 | 67.92±0.19 | 67.06±0.53 | 67.19±0.33 | 67.28±0.55 | 67.86±0.26 | **68.09±0.34** | 68.04±0.54 |
| | | AUROC | 83.72±0.38 | 84.67±0.18 | 83.84±0.47 | 84.13±0.39 | 84.33±0.22 | 84.77±0.06 | 84.72±0.26 | **84.98±0.09** |
| | | AUPRC | 73.33±0.68 | **75.29±0.16** | 73.23±0.99 | 73.57±0.71 | 74.18±0.53 | 75.07±0.24 | 74.32±0.41 | 74.78±0.23 |
| | Readmission | F1 | 83.79±0.11 | 83.82±0.08 | 83.71±0.28 | 83.52±0.16 | 84.02±0.14 | 83.80±0.06 | 84.12±0.17 | **84.18±0.08** |
| | | AUROC | 70.15±0.84 | 71.45±0.23 | 70.89±0.39 | 70.48±0.66 | 71.20±0.16 | 70.94±0.36 | **72.12±0.18** | 72.08±0.25 |
| | | AUPRC | 83.61±0.65 | 84.24±0.15 | 84.19±0.23 | 83.84±0.45 | 84.28±0.06 | 83.86±0.28 | 84.70±0.14 | **84.85±0.14** |

## 4.4 CASE STUDY

Beyond strong predictive performance, DT-BEHRT offers enhanced interpretability: its DA and DP modules mirror physicians' reasoning by focusing on disease groups and their progression over time rather than scattered attention across lengthy code sequences. We demonstrate this advantage through case studies on the MIMIC-IV phenotyping prediction task.

*Case 1 (Subject ID: 10253803, male, 59 years old; Figure 3)*: The patient had three hospital visits. In the subsequent visit, diagnoses included chronic obstructive pulmonary disease, congestive heart failure, other lower respiratory disease, and pneumonia. The DA module captured the relevance of existing respiratory conditions: within ICD-9 Chapter 460–519 (Diseases of the Respiratory System), codes such as 496 (chronic airway obstruction) and 491.21 (obstructive chronic bronchitis with acute exacerbation) received higher attention, whereas short-term symptoms or complications like 511.9 (unspecified pleural effusion) and 518.0 (pulmonary collapse) were assigned lower weights. The DP module highlighted cardiovascular progression across visits, from V45.81 (history of coronary artery bypass graft) in the first visit to 414.00 (coronary atherosclerosis) in the subsequent two visits, forming a clinically coherent trajectory. Finally, in the PR module, the most recent DP token received the highest attention, indicating that the model effectively leveraged temporal disease progression patterns. An additional case study can be found in Appendix I.

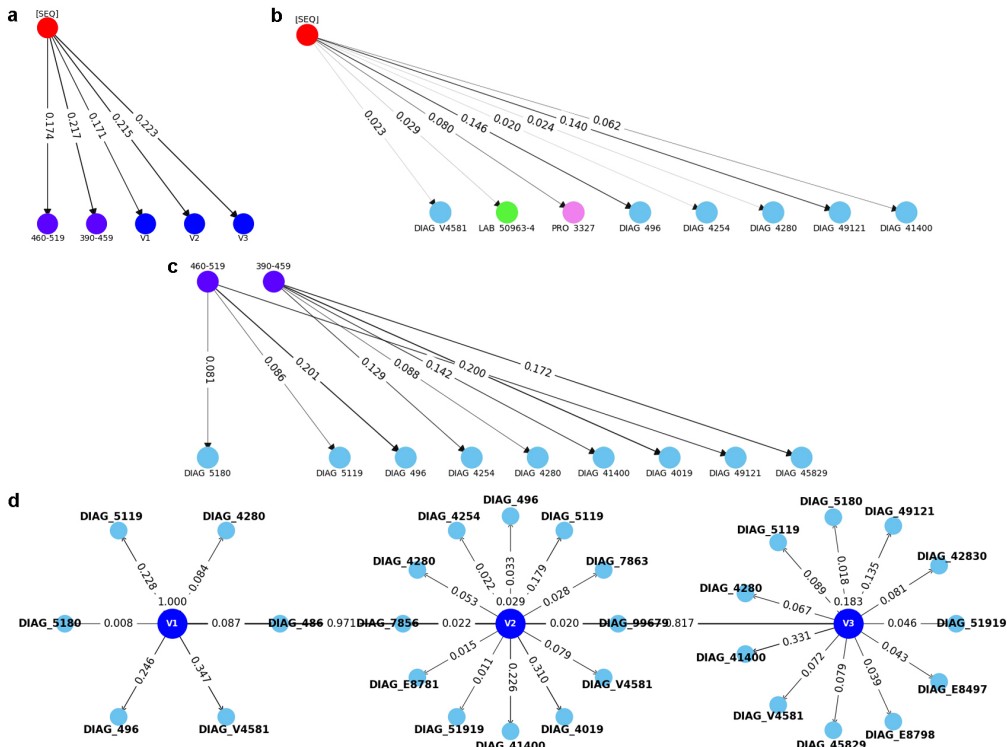

Figure 3: Illustration of Case 1 with attention scores of the **a** PR module, **b** SR module, **c** DA module, and **d** DP module. Only edges with scores > 0.001 are displayed, and self-loops are removed.

## 5 DISCUSSION

In this work, we present DT-BEHRT, a disease trajectory-aware transformer that integrates graph-enhanced modules into a sequential modeling framework. By explicitly centering diagnosis codes and modeling their progression and interactions across visits, DT-BEHRT addresses limitations of prior sequence-based and graph-based approaches that treat heterogeneous medical codes uniformly. We further design a tailored pre-training strategy combining global trajectory-level code masking and ontology-informed ancestor prediction, which encourages alignment across architectural components and improves the robustness of learned patient representations.

Across three benchmark EHR datasets, DT-BEHRT demonstrates competitive performance. Improvements are most notable for readmission prediction in MIMIC-III and for phenotyping prediction among patients with multiple hospital visits. Subgroup analyses further indicate that the benefits of its design are not limited to specific patient populations. In addition, case studies illustrate how the DA and DP modules highlight clinically coherent patterns, providing interpretability by aligning with common diagnostic reasoning processes.

While DT-BEHRT demonstrates strong performance, several limitations should be noted. First, the use of multi-head self-attention and graph attention across disease and visit nodes increases computational overhead and may limit scalability in resource-constrained settings. Second, the utility of the disease progression module is contingent on the presence of longitudinal trajectories. However, in the MIMIC datasets a substantial portion of patients have only one hospital visit, resulting in a degenerate graph structure with no temporal edges. Third, although this work is among the first to argue that different types of medical codes should be modeled differently, our design focuses primarily on diagnosis codes. Other code categories—such as medications, procedures, and laboratory tests—may also benefit from dedicated modeling structures, and exploring tailored mechanisms for these code types is an important direction for future work.

## REPRODUCIBILITY STATEMENT

The dataset used in this study is available on PhysioNet (https://physionet.org/), and the source code is publicly accessible at https://anonymous.4open.science/r/DT-BEHRT-C80F/README.md.

## LARGE LANGUAGE MODEL USAGE STATEMENT

Large language models were employed to support this work in limited ways. They were used for (i) literature search assistance, (ii) code debugging support, and (iii) grammar checking and language refinement of the manuscript. LLMs were not involved in research ideation, study design, data analysis, or interpretation of results.

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

## A  DETAILED RELATED WORK

As noted in the Section 1, research on EHR-based predictive modeling can be broadly classified into three methodological categories: sequence-based approaches, graph-based approaches, and graph-enhanced sequence approaches. In what follows, we provide a focused yet non-exhaustive review of widely benchmarked studies within each category, with particular emphasis on those most relevant to our proposed framework.

Sequence-based approaches. Early work in this area leveraged recurrent neural architectures. RE-TAIN (Choi et al., 2016), Dipole (Ma et al., 2017), and StageNet (Gao et al., 2020) are representative early sequence-based models developed without employing transformer architectures. RETAIN (Choi et al., 2016) employs a two-level attention mechanism to identify influential past visits and salient clinical variables within those visits. Dipole (Ma et al., 2017) leverages bidirectional recurrent neural networks to capture information from both past and future visits, while introducing attention mechanisms to quantify inter-visit relationships for prediction. StageNet (Gao et al., 2020) incorporates a stage-aware long short-term memory (LSTM; Hochreiter & Schmidhuber 1997) module to extract health stage variations in an unsupervised manner, along with a stage-adaptive convolutional module to integrate stage-specific progression patterns into risk prediction.

With the emergence of transformers (Vaswani et al., 2017), BERT-style models quickly surpassed the performance of these earlier approaches. BEHRT (Li et al., 2020) adapts the transformer architecture to represent longitudinal patient records, treating medical codes as tokens and temporal ordering as positional embeddings, thereby capturing long-range dependencies in patient trajectories. Med-BERT (Rasmy et al., 2021) scales pretraining to millions of patient records, enabling robust contextual embeddings of medical codes that can be fine-tuned for a wide range of downstream clinical prediction tasks. CEHR-BERT (Pang et al., 2021) incorporates temporal information through a hybrid strategy that augments the input with artificial time tokens, integrates time, age, and concept embeddings, and introduces an auxiliary learning objective for visit type prediction. TransformEHR (Yang et al., 2023) departs from the encoder-only paradigm by adopting an encoder–decoder framework and designing novel pretraining objectives to enhance performance. ExBEHRT (Rupp et al., 2023) extends the feature space to multimodal records by unifying the frequency and temporal dimensions of heterogeneous features, thereby facilitating comprehensive patient representation. Collectively, these transformer-based approaches significantly advance the state of the art by leveraging large-scale pretraining and contextualized representation learning to outperform prior sequence models.

Graph-based approaches. Graph-based modeling can be further categorized according to the underlying graph structure, including homogeneous graphs, heterogeneous graphs, and hypergraphs. Homogeneous graphs provide a relatively limited design space, as all nodes and edges share the same type. Consequently, they are often employed at the patient level rather than the code level. DEPOT (Song et al., 2023) exemplifies this line of work by constructing a patient similarity graph using $k$-nearest neighbors based on demographic features such as age and subsequently learning patient representations for prediction.

In contrast, heterogeneous graphs offer a higher modeling resolution at the code level, as they are more expressive than homogeneous graphs. HSGNN (Liu et al., 2020) and TRANS (Chen et al., 2024) represent recent advances in this subfield. HSGNN (Liu et al., 2020) decomposes a global EHR heterogeneous graph—consisting of medical code nodes, visit nodes, and patient nodes—into subgraphs defined by meta-paths, which are then fed into an end-to-end model for prediction. TRANS (Chen et al., 2024) constructs a temporal heterogeneous graph and explicitly encodes temporal information on edges to facilitate the propagation of temporal relationships.

Using hypergraphs to model EHR data is a relatively new direction. Unlike pairwise graphs, hypergraphs naturally capture higher-order interactions by allowing a hyperedge to connect multiple nodes. HCL (Cai et al., 2022) jointly learns patient embeddings and code embeddings by leveraging patient–patient, code–code, and patient–code relationships, while incorporating contrastive learning to enhance representation quality. Similarly, HypEHR (Xu et al., 2023) employs a hypergraph neural network, with SetGNN (Chien et al., 2021) as the backbone, to learn visit-level representations through high-order interactions.

Graph-enhanced sequence approaches. To combine the strengths of both sequence-based and graph-based modeling, a growing line of work has focused on graph-enhanced sequence models. At the medical code level, GRAM (Choi et al., 2017) enriches code embeddings with hierarchical information inherent in medical ontologies, which are represented as a knowledge-directed acyclic graph. KAME (Ma et al., 2018) not only learns meaningful embeddings for nodes in the knowledge graph but also leverages external knowledge through a knowledge-attention mechanism to improve prediction accuracy. Similarly, G-BERT (Shang et al., 2019) incorporates graph neural networks to represent the hierarchical structures of medical codes, and integrates these graph-based embeddings into a transformer-based visit encoder. The model is then pretrained on EHR data to capture contextualized code representations.

Moving beyond the code level, GCT (Choi et al., 2020) is a pioneering work that applies graph modeling at the visit level. It employs masked self-attention to learn a latent medical code graph within a visit and regularizes attention scores to mimic real-world co-occurrence patterns. However, temporal dependencies across visits are only weakly modeled. TPGT (Hadizadeh Moghaddam et al., 2025) and DeepJ (Li et al., 2025) extend GCT (Choi et al., 2020) by enhancing temporal awareness across visits. More recently, GT-BEHRT (Poulain & Beheshti, 2024) combines an architecture inspired by GCT (Choi et al., 2020) with a novel pretraining framework to further improve predictive performance.

At the patient level, Pellegrini et al. (2023) adopts a Graphormer (Ying et al., 2021) backbone to integrate heterogeneous, multimodal clinical data into population-level graphs, enabling unsupervised patient outcome prediction at scale. In parallel, HEART (Huang et al., 2024) introduces modified GAT layers to facilitate message passing across multiple visits of the same patient, thereby modeling longitudinal dependencies more effectively. It also models code heterogeneity primarily by augmenting code-type embeddings within a single attention-based aggregation, without introducing architectural-level heterogeneity.

# B  NOTATION TABLE

Table 4: Notations used in this paper.

| Notation | Description |
|---|---|
| $c, \mathcal{C}$ | A medical code; the medical code vocabulary |
| $\mathcal{D}, \mathcal{M}, \mathcal{L}, \mathcal{P}$ | Sets of diagnosis, medication, laboratory test, and procedure codes |
| $T$ | Total number of hospital visits |
| $v_t, \mathcal{V}$ | Set of codes at visit $t$; the entire sequence of visits / patient trajectory |
| $N_{v_t}, N_V$ | Number of codes in visit $v_t$ and total number of codes in trajectory $\mathcal{V}$, i.e., $N_V = \sum_{t=1}^{T} N_{v_t}$ |
| $\boldsymbol{e}_c, \boldsymbol{e}_{type(c)}, \boldsymbol{e}_{visit(c)}$ | Embedding vector of code $c$, its type, and its visit index |
| $L$ | Total number of hidden layers |
| $L_{\mathcal{G}}$ | Total Number of GAT blocks |
| $\boldsymbol{h}_c^{(l)}$ | Hidden representation vector at the $l$-th layer for code $c$ |
| $\boldsymbol{H}^{(l)} \in \mathbb{R}^{(1+N_V) \times \mathtt{d}}$ | Hidden representation matrix at the $l$-th layer |
| $\mathcal{J}, j$ | Index set of top-level ICD-9 categories; a top-level category index |
| $a_j$ | The $j$-th ICD-9 ancestor category |
| $\mathcal{D}_j$ | Diagnosis codes in category $j$ |
| $k$ | Threshold hyperparameter for triggering a DA token |
| $\boldsymbol{a}_{\mathcal{V}}$ | Ordered DA-token vector for a patient trajectory $\mathcal{V}$ |
| $N_a$ | Number of DA-tokens in $\boldsymbol{a}_{\mathcal{V}}$, i.e., $N_a = |\boldsymbol{a}_{\mathcal{V}}|$ |
| $\phi(l)$ | Category index of the DA token at row $l$ of attention mask ($l > 1 + N_V$) |
| $\boldsymbol{M} \in \mathbb{R}^{(1+N_V+N_a)^2}$ | Attention mask |
| $\boldsymbol{Z} \in \mathbb{R}^{N_a \times \mathtt{d}}$ | Representation set of DA tokens at the last layer |
| $\mathtt{d}$ | Hidden representation dimension |
| $\alpha$ | Masking rate |
| $\mathcal{A}$ | Global set (inventory) of DA tokens |
| $\mathrm{Anc} : \mathcal{D} \to \mathcal{J}$ | Ancestor-category map for diagnosis codes |
| $\mathrm{anc}_{\mathcal{V}}(j)$ | Number of distinct codes from $\mathcal{D}_j$ that appear in trajectory $\mathcal{V}$ |
| $[\mathrm{SEQ}]$ | Special sequence token |
| $\boldsymbol{V}$ | The visit-major vector prepended with $[\mathrm{SEQ}]$ flattened from $\mathcal{V}$ |
| $\boldsymbol{V}_{\boldsymbol{a}} = [\boldsymbol{V} \,||\, \boldsymbol{a}_{\mathcal{V}}]$ | Final concatenated sequence of length $1 + N_V + N_a$ |
| $\mathcal{G} = (\mathcal{U}, \mathcal{E}, \mathcal{X})$ | DP graph; node set; edge set; node-feature set |
| $\tilde{v}_t$ | DP visit node for visit $t$ |
| $d_{t,i}$ | The $i$-th diagnosis code in visit $t$ |
| $\tilde{d}_{t,i}$ | The $i$-th diagnosis node connected to the $t$-th DP node |
| $N_{d_t}$ | Number of diagnosis codes in visit $t$ |
| $\tau_{\mathcal{D}}, \tau_{\mathcal{M}}, \tau_{\mathcal{L}}, \tau_{\mathcal{P}}$ | Task type corresponding to the code sets $\mathcal{D}, \mathcal{M}, \mathcal{L}, \mathcal{P}$ |
| $\sigma(\cdot)$ | Sigmoid activation function |
| $Y_{\mathrm{mask}, \tau}$ | Masked token label for code type $\tau$ |
| $\ell_{\mathrm{anc}}, \ell_{\mathrm{anc,SR}}, \ell_{\mathrm{anc,DP}}; \lambda_{\mathrm{anc}}$ | Ancestor diagnosis code prediction losses (overall / SR / DP); penalizing coefficient |
| $\ell_{\mathrm{mask}}, \lambda_{\mathrm{mask}}$ | Masked token prediction loss; its weight |
| $\ell_{\mathrm{cov}}, \lambda_{\mathrm{cov}}$ | DA decorrelation penalty; its weight |
| $\ell_{\mathrm{task}}, \ell_{\mathrm{pt}}, \ell_{\mathrm{ft}}$ | Binary prediction, pre-training, and fine-tuning losses |

## C ICD-9 TOP-LEVEL CHAPTERS

Table 5: Nineteen top-level ICD-9 chapters and their code ranges.

| Code Range | ICD-9-CM Chapters |
|---|---|
| 001-139 | Infectious and parasitic diseases |
| 140-239 | Neoplasms |
| 240-279 | Endocrine, nutritional and metabolic diseases, and immunity disorders |
| 280-289 | Diseases of the blood and blood-forming organs |
| 290-319 | Mental, behavioral and neurodevelopmental disorders |
| 320-389 | Diseases of the nervous system and sense organs |
| 390-459 | Diseases of the circulatory system |
| 460-519 | Diseases of the respiratory system |
| 520-579 | Diseases of the digestive system |
| 580-629 | Diseases of the genitourinary system |
| 630-679 | Complications of pregnancy, childbirth, and the puerperium |
| 680-709 | Diseases of the skin and subcutaneous tissue |
| 710-739 | Diseases of the musculoskeletal system and connective tissue |
| 740-759 | Congenital anomalies |
| 760-779 | Certain conditions originating in the perinatal period |
| 780-799 | Symptoms, signs, and ill-defined conditions |
| 800-999 | Injury and poisoning |
| E000-E999 | Supplementary classification of external causes of injury and poisoning |
| V01-V91 | Supplementary classification of factors influencing health status and contact with health services |

## D SAMPLE ATTENTION MASK

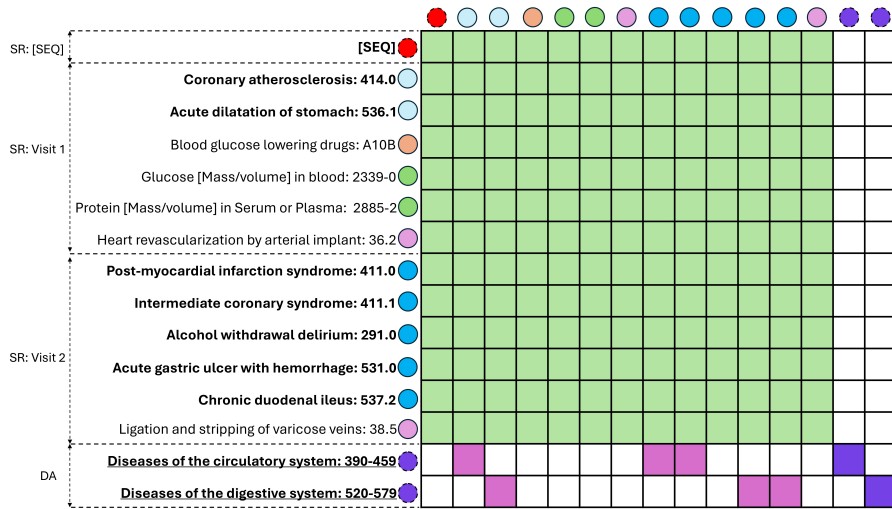

Figure 4: A sample attention mask with a DA token triggering threshold of $k = 3$. Each DA token is restricted to attend only to diagnosis codes within its corresponding ICD-9 chapter and to itself. In this case, ICD-9 code 291.0 (Alcohol withdrawal delirium) does not trigger a DA token.

## E    DATA PREPROCESSING DETAILS

We utilize two publicly available single-site EHR databases, MIMIC-III and MIMIC-IV, which contain records of patients admitted to the Beth Israel Deaconess Medical Center. Both datasets are structured hierarchically: each patient record consists of multiple hospital visits, and each visit includes diverse entities such as age, diagnoses, procedures, medications, and laboratory test results. For both MIMIC-III and MIMIC-IV, we applied the same preprocessing pipeline. Patient visits were arranged in chronological order, and for each visit, we extracted ICD-9 codes for diagnoses and procedures, NDC codes for medications, and item IDs for laboratory tests. Medications prescribed within the first 24 hours of a visit were retained, and NDC codes were subsequently mapped to ATC codes. We normalized age values greater than 90 to 90, and discretized the overall age range into 20 evenly distributed bins. For laboratory tests, numerical results were quantized into five categories by default, whereas categorical results were kept unchanged. Finally, we applied frequency-based filtering to reduce sparsity: only diagnoses appearing more than 2,000 times, procedures more than 800 times, and laboratory tests more than 1,500 times were retained. We also utilize a multi-site ICU database, eICU, following similar preprocessing procedures. The overall preprocessing strategy was consistent with that adopted in the HEART study (Huang et al., 2024). The statistics of the datasets are described in Table 6.

Table 6: Descriptive statistics of the MIMIC-III, MIMIC-IV, and eICU datasets.

| Dataset characteristics | MIMIC-III | MIMIC-IV | eICU |
|---|---|---|---|
| Number of patients | 33,067 | 60,709 | 85,839 |
| Diagnosis vocabulary size | 1,998 | 1,983 | 838 |
| Medication vocabulary size | 145 | 140 | 2,042 |
| Procedure vocabulary size | 801 | 801 | / |
| Laboratory test vocabulary size | 1,500 | 1,281 | 755 |
| Average visits per patient | 1.21 | 1.39 | 1.16 |
| Average diagnoses per visit | 10.83 | 10.26 | 3.31 |
| Average medications per visit | 7.82 | 2.98 | 24.12 |
| Average procedures per visit | 4.48 | 2.87 | / |
| Average laboratory tests per visit | 41.87 | 15.08 | 39.92 |
| In-hospital mortality (%) | 26.85 | 9.08 | 32.84 |
| Prolonged hospital stay (%) | 50.59 | 33.37 | 38.97 |
| Readmission rate (%) | 40.15 | 70.79 | / |

## F    IMPLEMENTATION DETAILS

For each of the baselines and our model, we perform 5 random runs and report the mean and standard deviation of test performance. The reported results correspond to the best model on the validation set, selected with an early stopping patience of 5. All experiments are conducted on a machine with a single NVIDIA A100 GPU (40GB memory). Our implementation is based on Python (3.10.18), PyTorch (1.13.1), and PyTorch Geometric (2.7.0). We adopt AdamW as the optimizer for all models. The hyperparameter search space of DT-BEHRT is summarized in Table 7.

Table 7: Model parameters and their search space.

| Parameters | Search Space |
|---|---|
| Learning rate | {0.01, 0.001} |
| Batch size | {32, 64} |
| Number of layers ($L$) | {2, 3} |
| Number of GAT blocks ($L_{\mathcal{G}}$) | {2} |
| Hidden representation dimension (d) | {64, 128} |
| Threshold for triggering a DA token ($k$) | {3, 4} |
| GCMP masking rate ($\alpha$) | {0.5, 0.6, 0.7} |
| Coefficient of the ACP loss ($\lambda_{\mathrm{anc}}$) | {0.05, 0.005} |
| Coefficient of the DA decorrelation loss ($\lambda_{\mathrm{cov}}$) | {0.05, 0.005} |

# G  PSEUDOCODE OF DT-BEHRT PRE-TRAINING AND FINE-TUNING

---

**Algorithm 1:** DT-BEHRT: Pre-training and Fine-tuning

---

**Input:** Hyperparameters $(epoch_{max}, L, \mathtt{d}, L_{\mathcal{G}}, k, \alpha, \lambda_{\text{anc}}, \lambda_{\text{cov}})$
**Output:** Trained parameters and patient representation $\boldsymbol{h}_{\text{[CLS]}}$
**Stage: Pre-training (GCMP + ACP)**
**Data:** Subset of patient trajectories $\mathcal{V}$ of medical codes $c \in \mathcal{C}$ for pre-training.

1 **Initialize** *model weights; optimizer.*
2 **for** $epoch = 1, \ldots, epoch_{max}$, **do**
3    **for** *mini-batch $\mathcal{B}$ of patients,* **do**
4       For each code type $\tau \in \mathcal{T}$, sample unique codes $Y_{\text{mask},\tau}$, at rate $\alpha$ and mask all occurrences
5       Initialize token embeddings $\boldsymbol{H}^{(0)}$ given in Equation 1; Initialize DP graph: visit nodes $\{\tilde{v}_t\}_{t=1}^T$ with embeddings $\boldsymbol{h}_{\tilde{v}_t}^{(0)} = \boldsymbol{e}_{Age(t)}$ and diagnosis nodes $\{\tilde{d}_{t,i}\}$ with embeddings $\boldsymbol{h}_{\tilde{d}_{i,t}}^{(0)} = \boldsymbol{h}_{d_{i,t}}^{(0)}$; Build chapter-restricted attention mask $\boldsymbol{M}$ via Equation 4
6       **for** $l = 1, \ldots, L$, **do**
7          Pass through pre-norm transformer layer in SR module $\ell$ via Equations 2-3 to get $\boldsymbol{H}^{(\ell)}$; Pass through a GAT layer in DP graph via Equations 6-8
8       Obtain the patient-level representation $\boldsymbol{h}_{\text{[CLS]}}$ via Equation 9
9       Predict masked codes with type-specific heads from $\boldsymbol{h}_{\text{[CLS]}}$ to get $\ell_{\text{mask}}$ as given in Equation 10
10      Predict ICD-9 chapter ancestors using $\boldsymbol{h}_{\text{[SEQ]}}^{(L)}$ and $\boldsymbol{h}_{\tilde{v}_T}^{(L)}$ to obtain $\ell_{\text{anc}} = \ell_{\text{anc,SR}} + \ell_{\text{anc,DP}}$ via Equations 11-12
11      Extract last-layer DA representations $\boldsymbol{Z}$ from $\boldsymbol{H}^{(L)}$ and compute de-correlation loss $\ell_{\text{cov}}$ via Equation 5
12      Form $\ell_{\text{pt}} = \ell_{\text{mask}} + \lambda_{\text{anc}}\ell_{\text{anc}} + \lambda_{\text{cov}}\ell_{\text{cov}}$ and update parameters by backprop on $\ell_{\text{pt}}$

**Return:** Pre-trained model weights
**Stage: Fine-tuning**
**Data:** Subset of Patient trajectories $\mathcal{V}$ for fine-tuning.
13 **Initialize** *with pre-trained model weights; optimizer.*
14 **for** *mini-batch* $(\mathcal{B}, Y_{\text{task}})$ **do**
15    Recompute $\boldsymbol{h}_{\text{[CLS]}}$ as in Steps 4-7 above
16    Compute task head prediction $\sigma(\text{Linear}(\boldsymbol{h}_{\text{[CLS]}}))$ and form $\ell_{\text{ft}} = \ell_{\text{task}} + \lambda_{\text{cov}}\ell_{\text{cov}}$
17    Update parameters by backprop on $\ell_{\text{ft}}$

**Return:** $\boldsymbol{h}_{\text{[CLS]}}$

---

# H  MEDICAL CODE REFERENCE TABLE

Table 8: Reference table of medical codes appearing in figures/text.

| Domain | Code | Code Type | Label |
|--------|------|-----------|-------|
| Diagnosis | 041.11 | ICD-9 | Methicillin susceptible Staphylococcus aureus in conditions classified elsewhere and of unspecified site |
| Diagnosis | 211.6 | ICD-9 | Benign neoplasm of pancreas, except islets of Langerhans |
| Diagnosis | 250.00 | ICD-9 | Diabetes mellitus without mention of complication, type II or unspecified type |
| Diagnosis | 250.40 | ICD-9 | Diabetes with renal manifestations, type II or unspecified type |
| Diagnosis | 250.50 | ICD-9 | Diabetes with ophthalmic manifestations, type II or unspecified type |
| Diagnosis | 250.60 | ICD-9 | Diabetes with neurological manifestations, type II or unspecified type |
| Diagnosis | 272.4 | ICD-9 | Other and unspecified hyperlipidemia |
| Diagnosis | 276.51 | ICD-9 | Dehydration |
| Diagnosis | 278.00 | ICD-9 | Obesity, unspecified |
| Diagnosis | 278.01 | ICD-9 | Morbid obesity |
| Diagnosis | 285.9 | ICD-9 | Anemia, unspecified |
| Diagnosis | 357.2 | ICD-9 | Polyneuropathy in diabetes |
| Diagnosis | 362.01 | ICD-9 | Background diabetic retinopathy |
| Diagnosis | 401.9 | ICD-9 | Unspecified essential hypertension |
| Diagnosis | 414.00 | ICD-9 | Coronary atherosclerosis of unspecified type of vessel |
| Diagnosis | 428.0 | ICD-9 | Congestive heart failure, unspecified |
| Diagnosis | 428.30 | ICD-9 | Diastolic heart failure, unspecified |
| Diagnosis | 425.4 | ICD-9 | Other primary cardiomyopathies |
| Diagnosis | 458.0 | ICD-9 | Orthostatic hypotension |
| Diagnosis | 458.1 | ICD-9 | Chronic hypotension |
| Diagnosis | 458.29 | ICD-9 | Other iatrogenic hypotension |
| Diagnosis | 486 | ICD-9 | Pneumonia, organism unspecified |
| Diagnosis | 491.21 | ICD-9 | Obstructive chronic bronchitis with acute exacerbation |
| Diagnosis | 496 | ICD-9 | Chronic airway obstruction, not elsewhere classified |
| Diagnosis | 511.9 | ICD-9 | Unspecified pleural effusion |
| Diagnosis | 518.0 | ICD-9 | Pulmonary collapse |
| Diagnosis | 519.19 | ICD-9 | Other diseases of trachea and bronchus |
| Diagnosis | 571.5 | ICD-9 | Cirrhosis of liver without mention of alcohol |
| Diagnosis | 571.8 | ICD-9 | Other chronic nonalcoholic liver disease |
| Diagnosis | 583.81 | ICD-9 | Nephritis and nephropathy, not specified as acute or chronic, in diseases classified elsewhere |
| Diagnosis | 585.9 | ICD-9 | Chronic kidney disease, unspecified |
| Diagnosis | 682.2 | ICD-9 | Cellulitis and abscess of trunk |
| Diagnosis | 785.6 | ICD-9 | Enlargement of lymph nodes |
| Diagnosis | 786.3 | ICD-9 | Hemoptysis |
| Diagnosis | 787.91 | ICD-9 | Diarrhea |
| Diagnosis | 810.02 | ICD-9 | Closed fracture of shaft of clavicle |
| Diagnosis | 996.79 | ICD-9 | Other complications due to other internal prosthetic device, implant, and graft |
| Diagnosis | 998.59 | ICD-9 | Other postoperative infection |
| Diagnosis | E849.7 | ICD-9 | Accidents occurring in residential institution |
| Diagnosis | E878.1 | ICD-9 | Surgical operation with implant of artificial internal device causing abnormal patient reaction, or later complication,without mention of misadventure at time of operation |
| Diagnosis | E878.8 | ICD-9 | Other specified surgical operations and procedures causing abnormal patient reaction, or later complication, without mention of misadventure at time of operation |
| Diagnosis | E885.9 | ICD-9 | Fall from other slipping, tripping, or stumbling |
| Diagnosis | E879.8 | ICD-9 | Other specified procedures as the cause of abnormal reaction of patient, or of later complication, without mention of misadventure at time of procedure |
| Diagnosis | V12.51 | ICD-9 | Personal history of venous thrombosis and embolism |
| Diagnosis | V45.81 | ICD-9 | History of coronary artery bypass graft |
| Diagnosis | V45.89 | ICD-9 | Other postprocedural status |
| Diagnosis | V58.61 | ICD-9 | Long-term (current) use of anticoagulants |
| Diagnosis | V58.67 | ICD-9 | Long-term (current) use of insulin |
| Diagnosis | V85.4 | ICD-9 | Body Mass Index 40 and over, adult |
| Lab | 54963-4 | LOINC | Diabetic foot ulcer(s) in last 7 days |
| Lab | 54082-3 | LOINC | Infectious diseases newborn screening panel |
| Medication | B01A | ATC | Antithrombotic agents |
| Medication | A04A | ATC | Antiemetics and antinauseants |
| Medication | N02A | ATC | Opioids |
| Medication | C03C | ATC | High-ceiling diuretics |
| Medication | C09A | ATC | ACE inhibitors, plain |
| Procedure | 33.27 | ICD-9 | Closed endoscopic biopsy of lung |
| Procedure | 52.59 | ICD-9 | Other and unspecified partial pancreatectomy |
| Procedure | 99.04 | ICD-9 | Transfusion of packed cells |
| Procedure | 41.5 | ICD-9 | Total splenectomy |

# I ADDITIONAL CASE STUDY

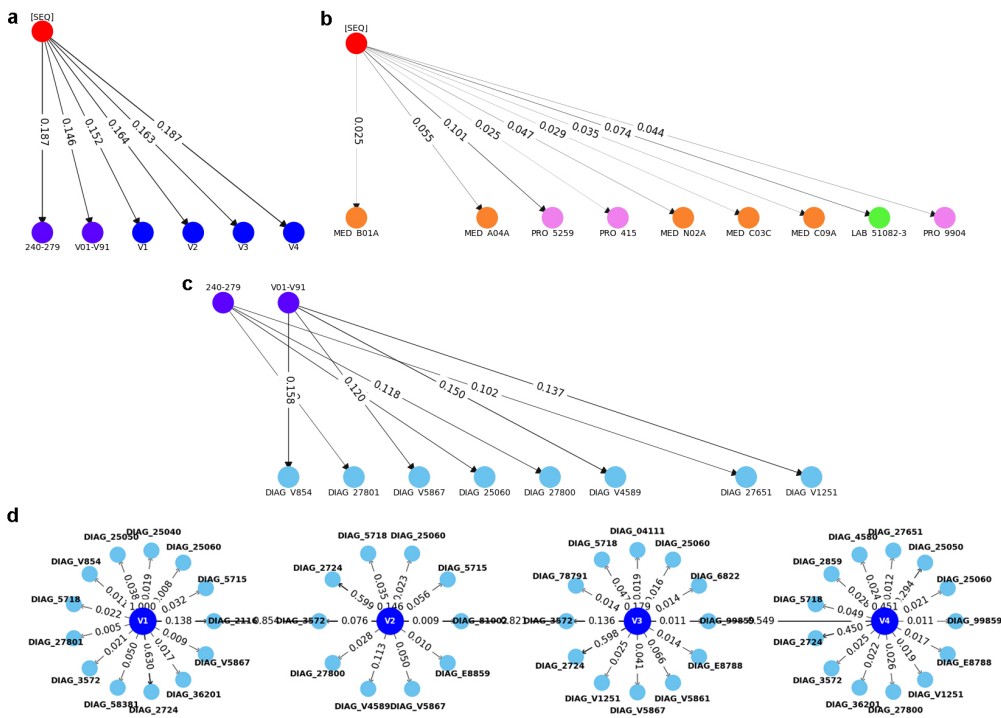

Figure 5: Illustration of Case 2 with attention scores of the **a** PR module, **b** SR module, **c** DA module, and **d** DP module. *Case 2 (Subject ID: 10725079, female, 63 years old).* The patient's subsequent diagnoses included Acute and unspecified renal failure, Cardiac dysrhythmias, Disorders of lipid metabolism, Fluid and electrolyte disorders, Gastrointestinal hemorrhage, and Septicemia (except in labor). In the PR module, we observe that codes within ICD-9 Chapter 240–279 (Endocrine, nutritional and metabolic diseases, and immunity disorders) received higher attention weights, which aligns with the patient's metabolic disorders likely secondary to renal failure. Furthermore, the attention assigned to DP tokens increased over time, indicating that the model captured the worsening trajectory of renal failure and its associated complications.

