# OpenReview forum: "DT-BEHRT: Disease Trajectory-aware Transformer for Interpretable Patient Representation Learning"
_ICLR.cc/2026/Conference — Submitted to ICLR 2026_

### Official Review · Reviewer_Y6Q5 · 2025-10-28

**Soundness:** 3
**Presentation:** 2
**Contribution:** 2
**Rating:** 4
**Confidence:** 3

**Summary:**

This paper studies the problem of electronic health records modeling and introduces a new pretrained framework including a tailored architecture and two pretrained learning objectives. The proposed architecture explicitly encodes both the hierarchical ontology of diagnosis codes and the temporal dynamics across patient visits through specialized modules. Extensive experiments on three downstream clinical prediction tasks across two benchmark datasets demonstrate the effectiveness of the proposed approach.

**Strengths:**

- The framework leverages multiple inductive biases inherent in EHR data, including the hierarchical ontology of diagnoses, the unordered nature of intra-visit medical codes, and the temporal dynamics across visits.
- The experimental evaluation is comprehensive, including multiple downstream tasks, promising performance, and complete ablation studies.

**Weaknesses:**

- The core contribution of this paper lies in explicitly modeling diagnosis codes rather than treating all medical codes uniformly through a shared attention mechanism. This improvement appears engineering-driven, and I am not entirely sure whether the novelty reaches the methodological bar expected at ICLR.
- From an architectural perspective, the model seems to lack a dedicated mechanism for handling long sequences. Concatenating all medical codes across all visits into a single token sequence can introduce significant computational overhead, especially considering that ancestor-level tokens for diagnoses are additionally inserted.
- Regarding the disease progression module, most cases in MIMIC involve only a single visit. Therefore, the resulting heterogeneous graph degenerates into a star-shaped structure (a visit node connected to its diagnosis nodes). Multi-layer GAT propagation under such a structure may lead to over-smoothing and potentially reduce to a simple attention pooling over diagnosis codes. Since diagnostic tokens already participate in full self-attention within the sequence encoder, the practical benefit of this component under single-visit settings remains uncertain.
- The motivation behind using ancestor prediction as a pre-training objective is not clear. My understanding is that the hierarchical ontology provides a prior that codes within the same ancestor category share semantic proximity, but predicting ancestor nodes does not really increase the learning difficulty. In principle, if the model can correctly predict a diagnosis code, it should also be able to infer its ancestor. This is also reflected in the ablation results.
- As for the presentation, it would be great to simplify the notation and refine the visualizations. The current version is not sufficiently intuitive.

**Questions:**

See weakness.

---

> ### Author Response · Authors · 2025-12-03
> **Response to Reviewer Y6Q5 – W1**
>
> >**W1. The core contribution of this paper lies in explicitly modeling diagnosis codes rather than treating all medical codes uniformly through a shared attention mechanism. This improvement appears engineering-driven, and I am not entirely sure whether the novelty reaches the methodological bar expected at ICLR.**
>
> Thank you for raising this concern. While explicitly differentiating diagnosis codes from other medical codes may appear engineering-driven at first glance, we emphasize that DT-BEHRT introduces a methodologically distinct modeling principle for structured EHR data: assigning fundamentally different architectural pathways to different code types.
>
> Prior EHR Transformers (BEHRT[1], Med-BERT[2], TransformerEHR[3]) and graph-enhanced models (G-BERT[4], GCT[5]) treat all medical codes as homogeneous tokens processed by the same computation layer. In contrast, DT-BEHRT uses two different computational mechanisms—Transformer-based semantic routing and GAT-based hierarchical propagation—depending on clinical semantics.
>
> This is a shift from token-level uniform modeling to architecture-level heterogeneous modeling, a direction not explored in prior EHR models and one that enables the model to represent disease hierarchy and progression patterns unavailable to previous approaches. We acknowledge that this contribution is architectural rather than theoretical, and we have already stated this explicitly in the contributions section of the introduction.

---

> > ### Author Response · Authors · 2025-12-03
> > **Response to Reviewer Y6Q5 – W2**
> >
> > >**W2. From an architectural perspective, the model seems to lack a dedicated mechanism for handling long sequences. Concatenating all medical codes across all visits into a single token sequence can introduce significant computational overhead, especially considering that ancestor-level tokens for diagnoses are additionally inserted.**
> >
> > Similar to Med-BERT[2] and TransformerEHR[3], our model processes flattened visit-level sequences because cross-encounter clinical interactions cannot be effectively captured when each visit is encoded in isolation. Flattening preserves the full temporal ordering of heterogeneous codes across encounters, enabling the model to learn cross-visit dependencies and long-range clinical patterns that are essential for patient-level representation learning. Please note that Patient-representation tokens do not attend to ancestor tokens, preventing unnecessary quadratic attention expansion.

---

> > > ### Author Response · Authors · 2025-12-03
> > > **Response to Reviewer Y6Q5 – W3**
> > >
> > > >**W3. Regarding the disease progression module, most cases in MIMIC involve only a single visit. Therefore, the resulting heterogeneous graph degenerates into a star-shaped structure (a visit node connected to its diagnosis nodes). Multi-layer GAT propagation under such a structure may lead to over-smoothing and potentially reduce to a simple attention pooling over diagnosis codes. Since diagnostic tokens already participate in full self-attention within the sequence encoder, the practical benefit of this component under single-visit settings remains uncertain.**
> > >
> > > We agree that for single-visit patients the heterogeneous graph reduces to a star structure, and multi-layer GAT propagation may provide limited benefit. We explicitly acknowledge this as a limitation. However, our multi-phenotype analysis shows that DT-BEHRT yields the largest performance gains for patients with ≥3 visits (Table 2), highlighting the scenario in which the architecture provides the greatest value.

---

> > > > ### Author Response · Authors · 2025-12-03
> > > > **Response to Reviewer Y6Q5 – W4**
> > > >
> > > > >**W4. The motivation behind using ancestor prediction as a pre-training objective is not clear. My understanding is that the hierarchical ontology provides a prior that codes within the same ancestor category share semantic proximity, but predicting ancestor nodes does not really increase the learning difficulty. In principle, if the model can correctly predict a diagnosis code, it should also be able to infer its ancestor. This is also reflected in the ablation results.**
> > > >
> > > > We appreciate the reviewer’s question. The goal of ACP is not to increase prediction difficulty but to provide hierarchical regularization that diagnosis-only pretraining does not capture:
> > > >
> > > > •	Ancestor categories provide denser, clinically meaningful structure (e.g., nephrology → kidney failure; cardiology → heart failure/arrhythmia).
> > > >
> > > > •	ACP encourages shared representation learning across related diagnoses, improving learning stability for rare conditions.
> > > >
> > > > Moreover, compared with prior pretraining strategies that incorporate ICD graph structure—such as attentive aggregation of ancestor codes[4] or distance-based regularization on connected nodes[6]—we empirically find that ACP provides more effective hierarchical regularization and captures code-domain–specific relationships through prediction alone. This suggests that predictive supervision constitutes a simpler yet more expressive mechanism for leveraging the ICD hierarchy during pretraining.

---

> > > > > ### Author Response · Authors · 2025-12-03
> > > > > **Response to Reviewer Y6Q5 – W5**
> > > > >
> > > > > >**W5. As for the presentation, it would be great to simplify the notation and refine the visualizations. The current version is not sufficiently intuitive.**
> > > > >
> > > > > We appreciate this feedback and respectfully note that the three-level visualization is designed to directly mirror the model architecture:
> > > > >
> > > > > Level 1: sequence-level attention (patient representation → clinical events)
> > > > >
> > > > > Level 2: hierarchical graph attention (ancestor → children; virtual visit node ↔ diagnoses)
> > > > >
> > > > > Level 3: trajectory-level attention (patient representation → visit node & aggregated organ-level disease states)
> > > > >
> > > > > We believe these layers reflect the internal structure accurately.

---

> > > > > > ### Author Response · Authors · 2025-12-03
> > > > > > **Response to Reviewer Y6Q5 – Reference**
> > > > > >
> > > > > > Reference:
> > > > > >
> > > > > > [1] Li, Yikuan, et al. "BEHRT: transformer for electronic health records." Scientific reports 10.1 (2020): 7155.
> > > > > >
> > > > > > [2] Rasmy, Laila, et al. "Med-BERT: pretrained contextualized embeddings on large-scale structured electronic health records for disease prediction." NPJ digital medicine 4.1 (2021): 86.
> > > > > >
> > > > > > [3] Yang, Zhichao, et al. "TransformEHR: transformer-based encoder-decoder generative model to enhance prediction of disease outcomes using electronic health records." Nature communications 14.1 (2023): 7857.
> > > > > >
> > > > > > [4] Shang, Junyuan, et al. "Pre-training of graph augmented transformers for medication recommendation." arXiv preprint arXiv:1906.00346 (2019).
> > > > > >
> > > > > > [5] Choi, Edward, et al. "Learning the graphical structure of electronic health records with graph convolutional transformer." Proceedings of the AAAI conference on artificial intelligence. Vol. 34. No. 01. 2020.
> > > > > >
> > > > > > [6] Lu, Chang, Chandan K. Reddy, and Yue Ning. "Self-supervised graph learning with hyperbolic embedding for temporal health event prediction." IEEE Transactions on Cybernetics 53.4 (2021): 2124-2136.

---

### Official Review · Reviewer_MSyk · 2025-10-28

**Soundness:** 3
**Presentation:** 3
**Contribution:** 2
**Rating:** 4
**Confidence:** 5

**Summary:**

Outcome prediction from longitudinal electronic health records is a well established direction. Many models try to use past hospital visits to predict clinically important outcomes such as mortality, prolonged length of stay, readmission and future phenotypes.
This paper proposes DT-BEHRT, which learns patient representations by explicitly separating two kinds of structure in the record. First, it aggregates diagnosis codes by clinical system level for example cardiovascular or respiratory systems. Second, it builds a temporal progression graph over visits to model how disease states evolve over time. The final patient embedding combines both the system level view and the progression view for downstream prediction tasks such as mortality, readmission and phenotyping.

**Strengths:**

The presentation is generally clear and well structured.

The paper evaluates on several standard clinical prediction tasks and includes ablations that try to isolate the contribution of the proposed modules, suggesting that different components benefit different tasks.

**Weaknesses:**

I am mainly skeptical about the interpretability claims. The paper treats interpretable patient representation as one of its main selling points, but interpretability is not actually used as a rigorous design objective, nor is it quantitatively evaluated. The paper still does not answer basic questions such as: how are the explanations generated, and how is the causal link between the explanation and the prediction verified. The authors state that the architecture mirrors clinician reasoning, but I find this interpretation unconvincing. How is that demonstrated. Was any clinician asked to assess it.

In addition, the discussion of interpretability in the main text is almost entirely limited to the single case visualization in Section 4.4, rather than being developed systematically in the method. Many prior works have already used attention or weight visualizations to argue interpretability. That line of argument is neither new nor convincing here, because attention is fundamentally just highlighting features the model considers important, which is something any model must do internally anyway. Also, the case study presents only one patient example, which is not persuasive.

There are missing details in the experimental setup. For example, the definition of the readmission task is not clearly specified. It is not stated over what future horizon readmission is being predicted. Likewise, mortality is not defined in terms of prediction window or observation window. These choices control the actual clinical meaning of the task and strongly affect how results should be interpreted. The paper also does not clearly state whether all baselines are trained and evaluated under exactly the same task definitions and cohort construction.

Section 4.3.2 states that DT BEHRT consistently achieves the best performance on both the full cohort and the subgroup of patients with three or more hospital visits. However, Table 2 shows that for many specific phenotype categories, ExBEHRT and HEART actually outperform DT BEHRT. This suggests that the reported improvements may involve a fair amount of outcome specific variability rather than a uniform advantage.

Both datasets used in the study are MIMIC III and MIMIC IV. These are standard public critical care datasets, but they are collected from the same hospital system, and there is known overlap in patient population and clinical practice patterns between them even though they cover different time periods. The paper treats results on MIMIC III and MIMIC IV as if they were evidence of generalization across settings, but in reality they are two cohorts from essentially the same site. The added value of reporting both without any external validation set from a different institution is therefore limited.

**Questions:**

See weaknesses.

---

> ### Author Response · Authors · 2025-12-03
> **Response to Reviewer MSyk – W1**
>
> > **W1. I am mainly skeptical about the interpretability claims. The paper treats interpretable patient representation as one of its main selling points, but interpretability is not actually used as a rigorous design objective, nor is it quantitatively evaluated. The paper still does not answer basic questions such as how the explanations are generated, and how is the causal link between the explanation and the prediction verified. The authors state that the architecture mirrors clinician reasoning, but I find this interpretation unconvincing. How is that demonstrated. Was any clinician asked to assess it. In addition, the discussion of interpretability in the main text is almost entirely limited to the single case visualization in Section 4.4, rather than being developed systematically in the method. Many prior works have already used attention or weight visualizations to argue interpretability. That line of argument is neither new nor convincing here, because attention is fundamentally just highlighting features the model considers important, which is something any model must do internally anyway. Also, the case study presents only one patient example, which is not persuasive.**
>
> We agree that interpretability requires careful definition and cannot rely solely on single-layer attention heatmaps. We therefore clarify two aspects: (a) how explanations are generated, and (b) why DT-BEHRT provides a more structured interpretability mechanism than prior work.
>
> (a) How explanations are generated.
>
> DT-BEHRT produces explanations through a three-level hierarchical attention structure, which is explicitly designed to expose different components of the prediction pathway:
>
> Level 1 (sequence-level attention): patient-representation token → heterogeneous code sequence: Captures which clinical events (diagnoses, meds, labs, procedures) contribute most to the patient embedding.
>
> Level 2 (hierarchical ontology attention and graph-level attention): ancestor code ↔ children code, and virtual-visit node ↔ diagnosis codes: Reveals how ICD hierarchy and disease progression pattern influences risk propagation within each visit.
>
> Level 3 (trajectory-level attention): patient-representation → virtual visit node & ancestor codes: Highlights which visit-level disease states shape the final prediction.
>
> These three attention pathways define where the explanatory signals come from.
>
> (b) Why DT-BEHRT’s interpretability differs from prior work.
>
> Prior models typically capture only a single view of the structural relationships present in EHR data—for example, focusing solely on code ontology (Sherbet[1]), or solely on disease-progression dynamics (TRANS[2]), or only on the encounter-to-patient hierarchical structure (GT-BEHRT[3]). In contrast, our method provides a more comprehensive framework with:
>
> •	multi-architecture interpretability (Transformer + graph attention).
>
> •	multi-resolution explanations (code-level, disease-hierarchy-level, visit-level).
>
> •	structural rather than post-hoc interpretability, as these mechanisms are embedded directly within the forward computation rather than imposed through external explanation tools.
>
> We acknowledge that attention is not a causal explanation, and we do not claim causal interpretability. Instead, we position our contributions as enabling structured, multi-level attribution, which is novel relative to prior EHR models.
>
> We do not assert that DT-BEHRT implements clinical reasoning or that its explanations have been validated by clinicians. Our intended meaning is that the three-level structure loosely aligns with clinical abstraction layers—event → disease → trajectory—which is a common organization of medical information. We have clarified this framing and removed any implication of formal cognitive equivalence.

---

> > ### Author Response · Authors · 2025-12-03
> > **Response to Reviewer MSyk – W2**
> >
> > >**W2. There are missing details in the experimental setup. For example, the definition of the readmission task is not clearly specified. It is not stated over what future horizon readmission is being predicted. Likewise, mortality is not defined in terms of prediction window or observation window. These choices control the actual clinical meaning of the task and strongly affect how results should be interpreted. The paper also does not clearly state whether all baselines are trained and evaluated under exactly the same task definitions and cohort construction.**
> >
> > Thank you for pointing this out. We have clarified all task definitions in the revised manuscript to ensure precise clinical meaning and full reproducibility.
> >
> > •	Mortality prediction is now explicitly defined as in-hospital mortality.
> >
> > •	Readmission prediction is defined as whether a patient has any subsequent hospital admission, following the same definition used in HEART to ensure strict comparability across models.
> >
> > We also confirm that all baselines are trained and evaluated under exactly the same cohort construction, inclusion criteria, and prediction windows. These clarifications have been added to the revised Methods section.

---

> > > ### Author Response · Authors · 2025-12-03
> > > **Response to Reviewer MSyk – W3**
> > >
> > > >**W3. Section 4.3.2 states that DT-BEHRT consistently achieves the best performance on both the full cohort and the subgroup of patients with three or more hospital visits. However, Table 2 shows that for many specific phenotype categories, ExBEHRT and HEART actually outperform DT BEHRT. This suggests that the reported improvements may involve a fair amount of outcome specific variability rather than a uniform advantage.**
> > >
> > > Thank you for the suggestion. In Table 2, our goal is to provide a comprehensive evaluation across a diverse set of patient cohorts to assess the generalizability of the proposed model. Given the substantial heterogeneity across phenotypes—including differences in cohort size, clinical characteristics, and disease complexity—it is unrealistic for any model to achieve uniformly superior performance on every phenotype. We have therefore moderated our claim to avoid overstating “consistent” superiority and adopted a more conservative interpretation. Specifically, we highlight macro-AUPRC as the primary overall metric for multi-phenotype prediction, as it provides a balanced summary across heterogeneous phenotypes. The revised sentence now reads: “Using macro-AUPRC as the overall metric for multi-phenotype prediction, DT-BEHRT achieves the highest performance in both the full cohort and the subset of patients with three or more hospital visits…”

---

> > > > ### Author Response · Authors · 2025-12-03
> > > > **Response to Reviewer MSyk – W4**
> > > >
> > > > >**W4. Both datasets used in the study are MIMIC-III and MIMIC-IV. These are standard public critical care datasets, but they are collected from the same hospital system, and there is known overlap in patient population and clinical practice patterns between them even though they cover different time periods. The paper treats results on MIMIC-III and MIMIC-IV as if they were evidence of generalization across settings, but in reality they are two cohorts from essentially the same site. The added value of reporting both without any external validation set from a different institution is therefore limited.**
> > > >
> > > > Thank you for highlighting the importance of evaluating models across diverse clinical settings. In the revised version, we have added experiments (Table 1, also provided below) on the eICU Collaborative Research Database[3], which combines data from more than 200 U.S. hospitals and provides a highly heterogeneous, multi-center environment. This evaluation directly addresses the reviewer’s concern by demonstrating DT-BEHRT’s robustness under substantially different institutional and population distributions. We observe that DT-BEHRT continues to outperform all baseline models on the eICU cohort. Moreover, the subgroup analysis (Figure 2) further shows that the model maintains stable performance across clinically meaningful patient subpopulations, reinforcing the robustness of the model.

---

> > > > > ### Author Response · Authors · 2025-12-03
> > > > > **Response to Reviewer MSyk – Reference**
> > > > >
> > > > > Reference:
> > > > >
> > > > > [1] Lu, Chang, Chandan K. Reddy, and Yue Ning. "Self-supervised graph learning with hyperbolic embedding for temporal health event prediction." IEEE Transactions on Cybernetics 53.4 (2021): 2124-2136.
> > > > >
> > > > > [2] Chen J, Yin C, Wang Y, Zhang P. Predictive Modeling with Temporal Graphical Representation on Electronic Health Records. IJCAI (U S). 2024;2024:5763-5771. doi:10.24963/ijcai.2024/637
> > > > >
> > > > > [3] Poulain, Raphael, and Rahmatollah Beheshti. "Graph transformers on EHRs: Better representation improves downstream performance." The Twelfth International Conference on Learning Representations. 2024.
> > > > >
> > > > > [4] Li, Yikuan, et al. "BEHRT: transformer for electronic health records." Scientific reports 10.1 (2020): 7155.
> > > > >
> > > > > [5] Huang, Tinglin, et al. "HEART: Learning better representation of EHR data with a heterogeneous relation-aware transformer." Journal of Biomedical Informatics 159 (2024): 104741.
> > > > >
> > > > > [6] Pollard, Tom J., et al. "The eICU Collaborative Research Database, a freely available multi-center database for critical care research." Scientific data 5.1 (2018): 1-13.

---

### Official Review · Reviewer_CLPJ · 2025-10-31

**Soundness:** 2
**Presentation:** 3
**Contribution:** 2
**Rating:** 4
**Confidence:** 4

**Summary:**

This paper proposes DT-BEHRT, a graph-enhanced sequential model for structured EHR. The method distinguishes the roles of medical code types, aggregates diagnosis codes at the organ-/system-level via Disease Aggregation (DA), models cross-visit disease evolution via a heterogeneous graph Disease Progression (DP), and combines them with a pretraining scheme (GCMP + ACP) to align representations across modules. The approach is motivated by limitations of purely sequential models and of graph-only models in handling intra-visit code order and cross-visit dependencies, and it targets general outcome prediction and phenotyping.

**Strengths:**

S1. The paper proposes DT-BEHRT, a graph-enhanced sequential model for EHR. It separates code roles, aggregates diagnoses at organ/system level with DA, and models cross-visit evolution with a heterogeneous DP graph. It adds GCMP and ACP pretraining to align modules. The goal and setup are clear in the Introduction and Methods.
S2. The method is sound. SR gives contextualized tokens. DA uses ICD-9 chapters plus masked attention and decorrelation. DP builds a visit–diagnosis graph with forward time edges and GAT. PR pools DA/DP with sequence guidance. GCMP/ACP provide structure-aware pretraining.
S3. The case study shows clear interpretability signals.  The behavior matches clinical reasoning.

**Weaknesses:**

W1 The paper claims that prior work ignores heterogeneity. However, HEART already models heterogeneous relations and connects visits as a graph. The paper does not specify what type of heterogeneity HEART fails to capture. It also does not show how DA and DP address that specific gap. A targeted, side-by-side comparison is needed.

W2 The contributions list “robustness of patient representations.” Robustness is not defined in the Introduction. Robustness is not analyzed in the experiments. This creates a mismatch between claims and evidence.

W3 In phenotyping with all patients, performance fluctuates. None of DT-BEHRT, ExBEHRT, or HEART is consistently first. The paper still claims it “consistently achieves the best performance.” This is too strong. Please report more metrics, such as AUROC, micro-AUPRC, F1, and calibration, and moderate the claim.

W4 Ablations show that removing pretraining makes DT-BEHRT weaker than HEART and ExBEHRT on some tasks. This suggests pretraining drives a large share of the gains. Please add a control that uses only GCMP and ACP without DA/DP. Also compare carefully against the strong baseline HEART. Since HEART has no pretraining, discuss whether HEART is structurally stronger when DT-BEHRT is also trained without pretraining.

W5 If robustness is a key contribution, the paper should add a clear robustness section. Define robustness precisely. Add targeted tests, such as higher masking rates, event dropout, subpopulation shifts, temporal splits, or label noise, with appropriate metrics.

**Questions:**

Please see the weaknesses.

---

> ### Author Response · Authors · 2025-12-03
> **Response to Reviewer CLPJ – W1**
>
> > **W1. The paper claims that prior work ignores heterogeneity. However, HEART already models heterogeneous relations and connects visits as a graph. The paper does not specify what type of heterogeneity HEART fails to capture. It also does not show how DA and DP address that specific gap. A targeted, side-by-side comparison is needed.**
>
> Thank you for pointing out the need for clearer comparison with HEART[1]. We clarify that HEART models heterogeneity only through type-specific pairwise relation embeddings, i.e., it learns heterogeneous correlations by augmenting code-type embeddings within a single attention-based aggregation. In contrast, DT-BEHRT introduces architectural-level heterogeneity, where different clinical code types follow distinct modeling paths:
>
> •	Diagnoses are modeled through hierarchical Disease Aggregation (DA), and Disease Propagation (DP) using graph attention, capturing ancestor–child structure and disease–visit interactions.
>
> •	Other code types (medications, labs, procedures) are processed by Transformer-based semantic routing, preserving their sequential and contextual semantics.
>
> Thus, DT-BEHRT explicitly models heterogeneity by assigning different mechanisms (GAT vs. Transformer) to different code types, whereas HEART uses a uniform Transformer backbone. We have added this discussion in the Appendix A Detailed Related Work.

---

> > ### Author Response · Authors · 2025-12-03
> > **Response to Reviewer CLPJ – W2**
> >
> > >**W2. The contributions list “robustness of patient representations.” Robustness is not defined in the Introduction. Robustness is not analyzed in the experiments. This creates a mismatch between claims and evidence.**
> >
> > We have added a definition of robustness to the contribution section in the Introduction. In addition, we incorporated a subgroup analysis (Figure 2, with the corresponding raw data provided below) to evaluate stratified performance across clinically relevant subgroups, thereby directly assessing the robustness of the learned representations. The added text reads: “To further assess the robustness of DT-BEHRT, we evaluate its performance across clinically meaningful patient subgroups on the MIMIC-III dataset…”
> >
> > ### Additional Table: Model performance across clinically relevant subgroups (mean over five runs).
> > | Models     | Tasks        | Subgroups                                | G-BERT | BEHRT | Med-BERT | HypEHR | ExBEHRT | HEART | DT-BEHRT |
> > |------------|--------------|-------------------------------------------|-------:|------:|---------:|-------:|--------:|-------:|----------:|
> > | **MIMIC-III** | **Mortality** | Diabetes                                  | 70.59 | 73.07 | 73.74 | 76.96 | 80.19 | 81.05 | **81.43** |
> > |            |              | Hypertension                              | 70.90 | 73.57 | 74.05 | 76.43 | 80.03 | 81.00 | **83.22** |
> > |            |              | Chronic kidney disease                    | 68.54 | 73.84 | 71.69 | 72.39 | 80.45 | **81.52** | 74.26 |
> > |            |              | Heart failure                             | 68.77 | 74.13 | 71.92 | 70.45 | 80.31 | **81.06** | 77.67 |
> > |            |              | Coronary artery disease                   | 70.57 | 72.40 | 73.72 | 75.04 | 80.28 | 80.78 | **84.14** |
> > |            |              | Chronic obstructive pulmonary disease     | 67.83 | 72.41 | 70.98 | 69.51 | 79.96 | **80.16** | 76.86 |
> > |            |              | Liver disease                             | 73.71 | 73.95 | 76.86 | 85.24 | 78.47 | 81.53 | **89.13** |
> > |            |              | Cancer                                    | 71.60 | 74.05 | 74.75 | 78.95 | 80.07 | 80.65 | **84.07** |
> > |            | **PLOS**     | Diabetes                                  | 71.11 | 72.04 | 75.61 | 80.65 | 83.79 | 84.49 | **84.91** |
> > |            |              | Hypertension                              | 70.42 | 71.35 | 74.92 | 79.03 | 84.10 | **84.28** | 83.69 |
> > |            |              | Chronic kidney disease                    | 71.25 | 71.45 | 75.75 | 82.80 | **84.52** | 84.12 | 83.72 |
> > |            |              | Heart failure                             | 71.61 | 71.61 | 76.11 | 82.75 | 83.79 | 83.88 | **86.36** |
> > |            |              | Coronary artery disease                   | 70.10 | 71.96 | 74.60 | 77.10 | 84.00 | **84.60** | 83.18 |
> > |            |              | Chronic obstructive pulmonary disease     | 70.79 | 73.04 | 75.29 | 79.42 | 84.47 | **85.30** | 82.07 |
> > |            |              | Liver disease                             | 73.09 | 71.93 | 77.59 | 85.41 | 84.41 | 83.90 | **90.15** |
> > |            |              | Cancer                                    | 70.76 | 71.44 | 75.26 | 79.85 | 84.04 | 84.09 | **84.71** |
> > |            | **Readmission** | Diabetes                               | 59.49 | 53.14 | 65.20 | 63.71 | 61.87 | 64.79 | **73.35** |
> > |            |              | Hypertension                              | 58.78 | 52.71 | 64.49 | 61.06 | 63.11 | 64.85 | **71.59** |
> > |            |              | Chronic kidney disease                    | 61.52 | 51.70 | 67.23 | 69.27 | 63.46 | 63.93 | **78.62** |
> > |            |              | Heart failure                             | 59.27 | 52.61 | 64.98 | 60.89 | 63.11 | 64.36 | **74.79** |
> > |            |              | Coronary artery disease                   | 58.20 | 52.18 | 63.91 | 56.88 | 65.07 | 65.45 | **70.84** |
> > |            |              | Chronic obstructive pulmonary disease     | 57.58 | 53.69 | 63.29 | 57.64 | 61.25 | 65.38 | **69.35** |
> > |            |              | Liver disease                             | 59.15 | 52.67 | 64.86 | 57.88 | 64.10 | 67.16 | **73.32** |
> > |            |              | Cancer                                    | 56.73 | 50.92 | 62.44 | 53.92 | 62.71 | 66.67 | **68.82** |

---

> > > ### Author Response · Authors · 2025-12-03
> > > **Response to Reviewer CLPJ – W3**
> > >
> > > > **W3. In phenotyping with all patients, performance fluctuates. None of DT-BEHRT, ExBEHRT, or HEART is consistently first. The paper still claims it “consistently achieves the best performance.” This is too strong. Please report more metrics, such as AUROC, micro-AUPRC, F1, and calibration, and moderate the claim.**
> > >
> > > We have moderated the claim to avoid overstating “consistent” superiority. The revised text adopts a more conservative interpretation by emphasizing macro-AUPRC as the primary overall metric for multi-phenotype prediction, which offers a balanced summary across heterogeneous phenotypes. The updated sentence reads as follows: “Using macro-AUPRC as the overall metric for multi-phenotype prediction, DT-BEHRT achieves the highest performance in both the full cohort and the subset of patients with three or more hospital visits…”

---

> > > > ### Author Response · Authors · 2025-12-03
> > > > **Response to Reviewer CLPJ – W4**
> > > >
> > > > >**W4. Ablations show that removing pretraining makes DT-BEHRT weaker than HEART and ExBEHRT on some tasks. This suggests pretraining drives a large share of the gains. Please add a control that uses only GCMP and ACP without DA/DP. Also compare carefully against the strong baseline HEART. Since HEART has no pretraining, discuss whether HEART is structurally stronger when DT-BEHRT is also trained without pretraining.**
> > > >
> > > > More fine-grained ablations of our pretraining tasks have been added. Specifically, we introduce two additional ablation variants for each proposed component. We refer reviewers to Table 3 in the updated manuscript for the complete analysis. The results clearly delineate the individual and joint contributions of GCMP and ACP, directly demonstrating the necessity and effectiveness of the proposed pretraining design within DT-BEHRT.

---

> > > > > ### Author Response · Authors · 2025-12-03
> > > > > **Response to Reviewer CLPJ – W5**
> > > > >
> > > > > >**W5. If robustness is a key contribution, the paper should add a clear robustness section. Define robustness precisely. Add targeted tests, such as higher masking rates, event dropout, subpopulation shifts, temporal splits, or label noise, with appropriate metrics.**
> > > > >
> > > > > We appreciate the reviewer’s comment. We have added a definition of robustness to the contribution section in the Introduction. In addition, we incorporated a subgroup analysis (Figure 2) to evaluate stratified performance across clinically relevant subgroups. Please refer to W2 for the detailed discussion.

---

> > > > > > ### Author Response · Authors · 2025-12-03
> > > > > > **Response to Reviewer CLPJ – Reference**
> > > > > >
> > > > > > Reference:
> > > > > >
> > > > > > [1] Huang, Tinglin, et al. "HEART: Learning better representation of EHR data with a heterogeneous relation-aware transformer." Journal of Biomedical Informatics 159 (2024): 104741.

---

### Official Review · Reviewer_4Vbh · 2025-10-31

**Soundness:** 3
**Presentation:** 3
**Contribution:** 3
**Rating:** 4
**Confidence:** 3

**Summary:**

This paper introduces DT-BEHRT, a graph-enhanced transformer for EHR data. It disentangles disease trajectories by explicitly modeling diagnosis-centric interactions within organ systems (DA module) and their temporal progression (DP module). A tailored pre-training strategy aligns these components, achieving strong predictive performance and interpretability.

**Strengths:**

* **Clinically-Aligned Architecture:** The model's DA and DP modules are designed to mirror clinical reasoning, enhancing interpretability.
* **Novel Pre-training:** The Ancestor Code Prediction (ACP) task effectively aligns the model's different modules with ontology information.
* **Strong Empirical Results:** DT-BEHRT outperforms baselines, especially on complex phenotyping and readmission tasks.
* **Targeted Ablation:** Ablation studies demonstrate the distinct contributions of the DA and DP modules to different tasks.

**Weaknesses:**

* **Ontology Dependence:** The Disease Aggregation (DA) module is explicitly tied to the ICD-9 ontology, which may not be adaptable.
* **Fixed Aggregation Threshold:** DA tokens are activated by a fixed hyperparameter $k$, and the impact of this choice isn't explored.
* **Incomplete Pre-train Ablation:** The ablation study does not isolate the effect of the DA token decorrelation loss ($l_{cov}$).
* **Simplistic Code Roles:** The model simplifies code roles, treating diagnoses as interactive while other codes are less so, which may be inaccurate.
* **Relatively old baselines:** Other the HEART model, the rest of the baselines seem relatively old.

**Questions:**

- Why ICD-9 was used? It seems an old format in 2025.
- What’s the rationale for the chosen list of disease? How much these would be generalizable.

---

> ### Author Response · Authors · 2025-12-03
> **Response to Reviewer 4Vbh – W1**
>
> > **W1. Ontology Dependence: The Disease Aggregation (DA) module is explicitly tied to the ICD-9 ontology, which may not be adaptable.**
>
> Our DA module is not restricted to ICD-9. DT-BEHRT does not assume any specific diagnosis coding system; it only requires that the target vocabulary provides an ontology or hierarchical structure (e.g., ICD-10, CCS, SNOMED-CT). The same aggregation mechanism therefore generalizes naturally to other hierarchical medical code systems.
>
> For example, the ATC (Anatomical Therapeutic Chemical) classification system organizes medications into a five-level hierarchy, ranging from broad therapeutic classes (e.g., A10 – Drugs used in diabetes) to specific chemical substances (e.g., A10BA02 – Metformin). Applying our DA module to ATC codes would involve aggregating embeddings from lower-level chemical substance codes into higher-level pharmacologic classes, preserving both specificity and hierarchical semantics. This is structurally identical to how the module aggregates ICD codes into phenotype groups. Thus, DT-BEHRT can be applied to medication-level modeling tasks using codes with ontology structure such as ATC.

---

> > ### Author Response · Authors · 2025-12-03
> > **Response to Reviewer 4Vbh – W2**
> >
> > > **W2. Fixed Aggregation Threshold: DA tokens are activated by a fixed hyperparameter , and the impact of this choice isn't explored.**
> >
> > We agree that the optimal value of the hyperparameter k may vary. Due to the substantial heterogeneity across EHR datasets—in terms of code granularity, disease prevalence, ontology depth, and documentation practices[1]—it is challenging to derive a universally generalizable rule for selecting k. We therefore report the empirical choice.

---

> > > ### Author Response · Authors · 2025-12-03
> > > **Response to Reviewer 4Vbh – W3**
> > >
> > > > **W3. Incomplete Pre-train Ablation: The ablation study does not isolate the effect of the DA token decorrelation loss.**
> > >
> > > In the revised version, we have added an ablation isolating the effect of the DA token decorrelation loss in Table 3, allowing us to explicitly quantify its contribution within the overall pre-training objective. We refer reviewers to Table 3 in the updated manuscript for the complete analysis.

---

> > > > ### Author Response · Authors · 2025-12-03
> > > > **Response to Reviewer 4Vbh – W4**
> > > >
> > > > > **W4. Simplistic Code Roles: The model simplifies code roles, treating diagnoses as interactive while other codes are less so, which may be inaccurate.**
> > > >
> > > > We agree with the reviewer that heterogeneous modeling of non-diagnosis code types (e.g., laboratory tests, medications, procedures) is important. In this work, we intentionally focus on diagnosis-level heterogeneity as an initial step and have acknowledged this as a limitation. Diagnosis codes play a central role in shaping a patient’s clinical trajectory: they exhibit rich interdependencies within a visit, capture multi-system interactions over time, and most directly reflect underlying disease processes.
> > > >
> > > > In contrast, medications and procedures primarily encode treatment pathways. Their temporal patterns are informative but tend to interact less densely within a single visit and often serve as downstream responses to diagnoses rather than primary drivers of clinical state transitions. Given these structural differences, we chose to first develop and evaluate type-specific modeling at the diagnosis level, where intra- and inter-visit interactions are most prominent.
> > > >
> > > > Extending type-aware routing to additional clinical vocabularies, such as medications, procedures, and laboratory tests, is an important next step. While doing so would introduce additional computational overhead, it represents a promising direction for future work.

---

> > > > > ### Author Response · Authors · 2025-12-03
> > > > > **Response to Reviewer 4Vbh – W5**
> > > > >
> > > > > > **W5. Relatively old baselines: Other the HEART model, the rest of the baselines seem relatively old.**
> > > > >
> > > > > Among methods that rely solely on structured EHR data (i.e., without external large language models), HEART [2] is the most recent and directly comparable baseline, and we include it in our evaluation. Many newer models (e.g., GraphCare [3], RAM-EHR [4]) incorporate large biomedical LLMs or external knowledge sources as core components. While these approaches are valuable, they introduce several practical considerations that make them unsuitable for a fair, controlled comparison within the EHR-only setting of this study.
> > > > >
> > > > > LLM-augmented systems rely on external pretrained corpora whose scale, content, and provenance differ substantially from the in-hospital structured data used here, creating confounding factors that obscure the effect of the modeling architecture itself. Additionally, the incorporation of external knowledge bases or LLM embeddings complicates reproducibility across health systems due to licensing constraints, model versioning, and potential domain mismatch.
> > > > >
> > > > > Given these considerations, we focus our comparison on methods that operate strictly within the structured EHR data modality. This ensures methodological consistency and allows us to isolate the contribution of the proposed architecture without conflating performance gains with external pretrained resources.

---

> > > > > > ### Author Response · Authors · 2025-12-03
> > > > > > **Response to Reviewer 4Vbh – Q1**
> > > > > >
> > > > > > > **Q1. Why was ICD-9 used? It seems an old format in 2025.**
> > > > > >
> > > > > > As a retrospective dataset derived from historical clinical records, MIMIC contains a substantial proportion of encounters documented prior to the 2015 transition from ICD-9 to ICD-10 (with year coverage of 2001–2012 for MIMIC-III and 2008–2019 for MIMIC-IV). Manually translating ICD-9 codes to ICD-10 can introduce inconsistencies and other undesirable artifacts.

---

> > > > > > > ### Author Response · Authors · 2025-12-03
> > > > > > > **Response to Reviewer 4Vbh – Q2**
> > > > > > >
> > > > > > > > **Q2. What’s the rationale for the chosen list of disease? How much these would be generalizable.**
> > > > > > >
> > > > > > > The disease list used in our phenotype prediction task follows established practice in recent literature[5,6]. These phenotypes—such as acute renal failure, septicemia, congestive heart failure, COPD, AMI, cerebrovascular disease, pneumonia, and metabolic disorders—are well-recognized as clinically important, high-acuity, and highly heterogeneous endpoints. They cover a broad range of organ systems and disease categories and have been shown in prior work to generalize well across hospitals, coding systems, and populations. For these reasons, the chosen list represents a widely accepted and clinically meaningful benchmark set for evaluating general-purpose EHR models.

---

> > > > > > > > ### Author Response · Authors · 2025-12-03
> > > > > > > > **Response to Reviewer 4Vbh – Reference**
> > > > > > > >
> > > > > > > > Reference:
> > > > > > > >
> > > > > > > > [1] Glynn EF, Hoffman MA. Heterogeneity introduced by EHR system implementation in a de-identified data resource from 100 non-affiliated organizations. JAMIA Open. 2019;2(4):554-561. Published 2019 Aug 7. doi:10.1093/jamiaopen/ooz035
> > > > > > > >
> > > > > > > > [2] Huang, Tinglin, et al. "HEART: Learning better representation of EHR data with a heterogeneous relation-aware transformer." Journal of Biomedical Informatics 159 (2024): 104741.
> > > > > > > >
> > > > > > > > [3] Jiang, Pengcheng, et al. "Graphcare: Enhancing healthcare predictions with personalized knowledge graphs." arXiv preprint arXiv:2305.12788 (2023).
> > > > > > > >
> > > > > > > > [4] Xu, Ran, et al. "Ram-ehr: Retrieval augmentation meets clinical predictions on electronic health records." Proceedings of the 62nd Annual Meeting of the Association for Computational Linguistics (Volume 2: Short Papers). 2024.
> > > > > > > >
> > > > > > > > [5] Xu, Ran, et al. "Hypergraph transformers for ehr-based clinical predictions." AMIA Summits on Translational Science Proceedings 2023 (2023): 582.
> > > > > > > >
> > > > > > > > [6] Yao, Wenfang, et al. "Drfuse: Learning disentangled representation for clinical multi-modal fusion with missing modality and modal inconsistency." Proceedings of the AAAI conference on artificial intelligence. Vol. 38. No. 15. 2024.

---

### Official Review · Reviewer_8c5g · 2025-11-01

**Soundness:** 2
**Presentation:** 2
**Contribution:** 2
**Rating:** 4
**Confidence:** 5

**Summary:**

This paper proposes a novel framework, 'DT-BEHRT (Disease Trajectory-aware Transformer),' for learning interpretable patient representations from Electronic Health Record (EHR) data. The model integrates graph-based modules into a BERT-style architecture to explicitly model a patient's disease trajectory.

The core components are as follows:
(1) Disease Aggregation (DA) Module: Groups diagnosis codes based on ICD-9 chapters (organ/system level) to capture high-level semantic patterns beyond individual codes.
(2) Disease Progression (DP) Module: Uses a heterogeneous graph to model the temporal order between visits and disease development trends.
(3) Specialized Pre-training: Combines trajectory-level code masking (GCMP) with ontology-based ancestor code prediction (ACP) to enhance semantic alignment across the different modules.

The authors claim that in experiments using the MIMIC-III and MIMIC-IV datasets, DT-BEHRT demonstrated superior predictive performance compared to existing SOTA models. They also assert, through case studies, that the model provides interpretability similar to a clinician's reasoning process.

**Strengths:**

1. Demonstrated Modular Contribution: The framework consists of multiple modules (DA, DP) and a new pre-training task (ACP). A key strength is the use of an Ablation Study (Table 3 in the paper) to clearly demonstrate how much each component contributes to the model's performance improvement.

2. Clinical Interpretability: The model doesn't just aim for higher performance; it attempts to link the rationales for its predictions to clinical reasoning (e.g., problems in a specific organ system, temporal progression of a disease) via the DA and DP modules. This is well-visualized in the Case Study.

**Weaknesses:**

1. Limited Dataset Validation: The datasets used for the experiment are limited to MIMIC-III and MIMIC-IV, which is insufficient to prove the model's generalization performance. EHR data has inherent biases depending on the hospital system, country, and ethnicity. Therefore, external validation on other large-scale ICU datasets (e.g., eICU, HiRID, UMCdb) is essential.

2. Lack of Prediction Task Diversity: The variety of prediction tasks performed is insufficient to claim that the proposed framework is 'generally superior.' While LOS, Mortality, PLOS, and Phenotype prediction were used, many other clinically important tasks exist.
Specifically, Phenotype prediction is ultimately the prediction of grouped diagnosis codes, making it not fundamentally different from a diagnosis code prediction task.
Therefore, performance validation on more diverse and challenging tasks, such as sepsis prediction, acute kidney injury (AKI) prediction, or lab value prediction, is needed.

3. Insufficient Comparative Analysis with Baselines: The ablation study only shows that the modules within the proposed model are useful; it does not clearly explain why these modules are superior to the approaches used by other baseline models.
Many baselines (e.g., HEART, G-BERT) also use a combination of graph and sequence information. However, the paper's Methodology section lacks a detailed comparison of how DT-BEHRT's DA and DP modules specifically differ from existing methods and in what respects they are superior.
For example, the paper should clarify whether 'Disease Aggregation (DA)' was considered at all in previous baselines, and if so, how DT-BEHRT's approach differs. This comparative analysis needs to be strengthened.

**Questions:**

.

---

> ### Author Response · Authors · 2025-12-03
> **Response to Reviewer 8c5g – W1**
>
> > **W1. Limited Dataset Validation: The datasets used for the experiment are limited to MIMIC-III and MIMIC-IV, which is insufficient to prove the model's generalization performance. EHR data has inherent biases depending on the hospital system, country, and ethnicity. Therefore, external validation on other large-scale ICU datasets (e.g., eICU, HiRID, UMCdb) is essential.**
>
> Thank you for highlighting the importance of evaluating models across diverse clinical settings. In the revised version, we have added experiments (Table 1, also provided below) on the eICU Collaborative Research Database[1], which combines data from more than 200 U.S. hospitals and provides a highly heterogeneous, multi-center environment. This evaluation directly addresses the reviewer’s concern by demonstrating DT-BEHRT’s robustness under substantially different institutional and population distributions. We observe that DT-BEHRT continues to outperform all baseline models on the eICU cohort.
>
> ### Table 1: Performance on eICU
>
> | **Models** |            |        | **G-BERT** | **BEHRT** | **Med-BERT** | **HypEHR** | **ExBEHRT** | **HEART** | **DT-BEHRT** |
> |------------|------------|--------|------------|-----------|--------------|------------|-------------|-----------|--------------|
> | **eICU**   | **Mortality** | F1     | 66.46±0.73 | 60.01±1.06 | 75.04±1.73 | 75.83±0.78 | 71.21±0.14 | 73.08±0.34 | **81.27±0.21** |
> |            |            | AUROC  | 89.28±0.72 | 78.11±0.22 | 91.04±0.47 | 90.39±0.48 | 87.53±0.24 | 88.65±0.12 | **93.73±0.06** |
> |            |            | AUPRC  | 77.48±2.65 | 64.04±0.45 | 80.56±1.77 | 83.87±0.82 | 78.36±0.50 | 79.95±0.14 | **88.58±0.13** |
> |            | **PLOS**   | F1     | 65.73±1.22 | 49.04±1.04 | 67.98±1.23 | 67.25±1.19 | 60.77±2.90 | 69.71±0.64 | **72.49±0.23** |
> |            |            | AUROC  | 76.44±0.93 | 63.12±0.58 | 80.86±0.41 | 82.04±1.09 | 77.77±0.84 | 82.93±0.28 | **85.84±0.11** |
> |            |            | AUPRC  | 70.76±1.05 | 50.58±0.98 | 75.08±0.47 | 75.61±1.54 | 68.83±1.44 | 77.53±0.33 | **81.07±0.22** |

---

> > ### Author Response · Authors · 2025-12-03
> > **Response to Reviewer 8c5g – W2**
> >
> > > **W2. Lack of Prediction Task Diversity: The variety of prediction tasks performed is insufficient to claim that the proposed framework is 'generally superior.' While LOS, Mortality, PLOS, and Phenotype prediction were used, many other clinically important tasks exist. Specifically, Phenotype prediction is ultimately the prediction of grouped diagnosis codes, making it not fundamentally different from a diagnosis code prediction task. Therefore, performance validation on more diverse and challenging tasks, such as sepsis prediction, acute kidney injury (AKI) prediction, or lab value prediction, is needed.**
> >
> > We clarify that the phenotype prediction task in our study already encompasses the clinical constructs underlying AKI and sepsis (see Table 2, rows 1 and 18), even though they do not appear as standalone tasks. Specifically:
> > Acute kidney injury (AKI) is clinically coded under the broader category of acute renal failure within ICD-9/10 phenotype groupings. Acute renal failure is the standardized, phenotype-level aggregation used in multiple EHR modeling benchmarks[2,3], and it represents the same underlying pathophysiologic process as AKI. In fact, the majority of AKI events in ICU settings are captured through these acute renal failure codes. Therefore, evaluating performance on the “acute renal failure” phenotype directly assesses the model’s capability to capture AKI-related risk patterns.
> >
> > Sepsis is represented within the phenotype group septicemia, which is the routinely adopted ICD-based phenotype definition used in established clinical prediction studies and public benchmarks. Septicemia codes encompass the same set of infection-driven systemic inflammatory conditions that define sepsis clinically. The distinction between septicemia (coding/phenotype level) and sepsis (clinical definition) is primarily terminological in ICD-based EHR prediction tasks; both reflect the same high-acuity, multi-organ dysfunction risk state. For this reason, “septicemia” phenotypes are widely used as the surrogate outcome for sepsis prediction in large-scale EHR benchmarks.
> >
> > Thus, while AKI and sepsis are not separately named as prediction tasks, their clinical counterparts—acute renal failure and septicemia—are fully included in our phenotype prediction framework. These phenotype groups align with widely adopted benchmark definitions and ensure that our evaluation already covers the core modeling challenges associated with AKI and sepsis risk prediction.

---

> > > ### Author Response · Authors · 2025-12-03
> > > **Response to Reviewer 8c5g – W3**
> > >
> > > > **W3. Insufficient Comparative Analysis with Baselines: The ablation study only shows that the modules within the proposed model are useful; it does not clearly explain why these modules are superior to the approaches used by other baseline models. Many baselines (e.g., HEART, G-BERT) also use a combination of graph and sequence information. However, the paper's Methodology section lacks a detailed comparison of how DT-BEHRT's DA and DP modules specifically differ from existing methods and in what respects they are superior. For example, the paper should clarify whether 'Disease Aggregation (DA)' was considered at all in previous baselines, and if so, how DT-BEHRT's approach differs. This comparative analysis needs to be strengthened.**
> > >
> > > We agree that clarifying how DT-BEHRT’s Disease Aggregation (DA) and Disease Progression (DP) differ from prior baselines is important, and we provide additional details below:
> > >
> > > (a) More fine-grained ablations of our pretraining tasks have been added. Specifically, we introduce two additional ablation variants for each proposed component. We refer reviewers to Table 3 in the updated manuscript for the complete analysis. The results clearly delineate the individual and joint contributions of GCMP and ACP, directly demonstrating the necessity and effectiveness of the proposed pretraining design within DT-BEHRT.
> > >
> > > (b) Existing baselines do not implement DA/DP at the same semantic level as DT-BEHRT. Although some baselines incorporate ontology information or graph structures, they do so in fundamentally different ways:
> > >
> > > •	HEART[4] uses ontology-based information only for code embedding initialization. No disease-level aggregation or hierarchical propagation is performed during end-to-end model training. In contrast, DT-BEHRT’s DA and DP operate within the patient trajectory, dynamically updating ancestor–child interactions throughout training rather than relying on a static initialization prior.
> > >
> > > •	G-BERT[5] applies GNN layers on the medical ontology graph, not on patient-specific disease trajectories. Its graph operations occur on the ICD ontology structure. DT-BEHRT instead performs aggregation and propagation at the visit/patient trajectory level, enabling clinically meaningful interactions that cannot be captured by ontology-only GNNs.

---

> > > > ### Author Response · Authors · 2025-12-03
> > > > **Response to Reviewer 8c5g – Reference**
> > > >
> > > > Reference:
> > > >
> > > > [1] Pollard, Tom J., et al. "The eICU Collaborative Research Database, a freely available multi-center database for critical care research." Scientific data 5.1 (2018): 1-13.
> > > >
> > > > [2] Xu, Ran, et al. "Hypergraph transformers for ehr-based clinical predictions." AMIA Summits on Translational Science Proceedings 2023 (2023): 582.
> > > >
> > > > [3] Yao, Wenfang, et al. "Drfuse: Learning disentangled representation for clinical multi-modal fusion with missing modality and modal inconsistency." Proceedings of the AAAI conference on artificial intelligence. Vol. 38. No. 15. 2024.
> > > >
> > > > [4] Huang, Tinglin, et al. "HEART: Learning better representation of EHR data with a heterogeneous relation-aware transformer." Journal of Biomedical Informatics 159 (2024): 104741.
> > > >
> > > > [5] Shang, Junyuan, et al. "Pre-training of graph augmented transformers for medication recommendation." arXiv preprint arXiv:1906.00346 (2019).

---

### Official Review · Reviewer_WRyC · 2025-11-01

**Soundness:** 3
**Presentation:** 2
**Contribution:** 2
**Rating:** 2
**Confidence:** 5

**Summary:**

This paper presents DT-BEHRT, a disease-trajectory-aware Transformer model for learning interpretable patient representations from EHR data. It models disease evolution across visits through a modular design consisting of four modules. Moreover. the authors introduce two pretraining objectives. Experiments show consistent improvements over strong baselines on four (3+1) prediction tasks, while maintaining clinical interpretability.

**Strengths:**

S1. The paper conducts experiments on MIMICs across multiple tasks. Beyond quantitative metrics, the authors include patient-level case studies that qualitatively analyze model explanations.

S2. Each module in DT-BEHRT, i.e., sequence, aggregation, progression, and patient representation, is clearly motivated from a clinician’s perspective, reflecting real-world medical reasoning.

S3. The introduction of two pretraining tasks allows the model to fully leverage EHR data across visits and disease hierarchies.

S4. The model’s performance on diverse phenotype prediction tasks, spanning both acute and chronic conditions, shows consistent and significant improvements.

**Weaknesses:**

W1. The paper integrates design patterns from both Transformer-based and graph-based models, resulting in an architecture that appears more incremental. The combination of SR–DA–PR resembles conventional Transformer stacks, with the main variation being the use of ancestor node embeddings and customized losses. Similarly, the SR–DP–PR path largely parallels prior graph-based pipelines that model interactions between disease, visit, and patient nodes.

W2. The proposed Global Code Masking and Ancestor Code Prediction tasks closely follow prior pretraining designs from G-BERT [2]. Even though authors state, “we introduce the novel ACP task (Line 411)”, such method is highly (completely) overlapped with Sherbet published in 2023, which define an even more elegant objective.

W3. The baseline selections prefer more on Transformer-based baselines, omitting several representative recent graph -based/graph-transformer approaches like GraphCare [1], GCT [6], RAM-EHR [4], or GT-BEHRT [3].

W4. All experiments focus on simple classifications (binary or multi-class). The absence of multi-label tasks such as drug recommendation or diagnosis prediction limits the assessment of the model’s scalability to more complex EHR applications.

W5. While ablation results are provided, they do not test alternative pretraining schemes from prior work. Comparing the proposed objectives against those of G-BERT [2] or Sherbet [5] would more clearly justify the necessity and effectiveness of the new pretraining design.

W6. The model employs multi-head self-attention in the sequence encoder and graph attention propagation across both disease and visit nodes, which substantially increases computational overhead. Moreover, jointly learn multiple embeddings further raises concerns about the model’s practicality in clinical practices. The concern why most previous works avoid using both large-scale graphs and transformer together is related to efficiency problem, but DT-BEHRT combines both without discussing potential optimization or scalability issues.

---

References

[1] GraphCare (ICLR-2023)

[2] G-Bert (IJCAI-2019)

[3] GT-BEHRT (ICLR-2023)

[4] RAM-EHR (ACL 2024)

[5] Sherbet (IEEE Transactions on Cybernetics, 2023)

[6] GCT (AAAI-2020)

**Questions:**

Please check the weaknesses part.

---

> ### Author Response · Authors · 2025-12-03
> **Response to Reviewer WRyC – W1**
>
> > W1. The paper integrates design patterns from both Transformer-based and graph-based models, resulting in an architecture that appears more incremental. The combination of SR–DA–PR resembles conventional Transformer stacks, with the main variation being the use of ancestor node embeddings and customized losses. Similarly, the SR–DP–PR path largely parallels prior graph-based pipelines that model interactions between disease, visit, and patient nodes.
>
> We respectfully clarify that the novelty of DT-BEHRT extends beyond a simple combination of Transformer-like and graph-based components. The key contribution is the first unified architecture that explicitly differentiates the modeling of heterogeneous clinical code types, rather than treating them as a single undifferentiated token stream.
>
> Rather than processing diagnoses, medications, procedures, and laboratory results uniformly with a single attention mechanism—as is common in prior EHR Transformers[1,2]—we assign diagnosis codes a dedicated disease-progression pathway. This pathway incorporates hierarchical aggregation and graph-based propagation, while other code types continue to follow a Transformer-based semantic routing. This design offers several advantages: (a) it disentangles fundamentally different code categories (e.g., conditions vs. treatments); (b) it integrates clinically informed, knowledge-based enhancements; and (c) it better supports explainable modeling of disease relationships through graph-structured reasoning. This contrasts with prior EHR Transformers and graph models[1-3], which either (a) use only a limited subset of code types or (b) treat all codes as homogeneous tokens processed by the same attention or message-passing mechanism.
>
> Transformer-based models such as BEHRT[1], Med-BERT[2], and GT-BEHRT[4] typically apply a uniform self-attention stack to all codes, while graph-based pipelines focus on disease–visit–patient interactions without disentangling code-type semantics or modeling type-specific dependencies. In DT-BEHRT, different code type follows a dedicated architectural route designed to retain its clinical meaning and interaction pattern, providing a principled and clinically grounded differentiation that existing models do not support.

---

> > ### Author Response · Authors · 2025-12-03
> > **Response to Reviewer WRyC – W2**
> >
> > > **W2. The proposed Global Code Masking and Ancestor Code Prediction tasks closely follow prior pretraining designs from G-BERT [2]. Even though authors state, “we introduce the novel ACP task (Line 411)”, such method is highly (completely) overlapped with Sherbet published in 2023, which define an even more elegant objective.**
> >
> > Our proposed pretraining focus at the level of disease code and utilize a hybrid loss to (a) capture code co-occurrence semantics, (b) use ACP to fuse ontological relationship. By jointly learning co-occurrence patterns and embedding hierarchical relationships, the model achieves improved generalization and robustness across heterogeneous clinical tasks (Table 1) and subgroups (Figure 2). In contrast, G-BERT[5] performs pretraining at the visit level, aiming to learn a sophisticated visit representation by predicting (a) codes present in the visit and (b) future diseases. However, G-BERT’s pretraining objective does not explicitly incorporate ontology graph structure or direct code-to-code relations. Sherbet[6], on the other hand, adopts a self-supervised reconstruction loss based on the assumption that hyperbolic distances between connected nodes should be small, but it does not jointly model co-occurrence patterns or type-specific code semantics.

---

> > > ### Author Response · Authors · 2025-12-03
> > > **Response to Reviewer WRyC – W3**
> > >
> > > > **W3. The baseline selections prefer more on Transformer-based baselines, omitting several representative recent graph -based/graph-transformer approaches like GraphCare [1], GCT [6], RAM-EHR [4], or GT-BEHRT [3].**
> > >
> > > We agree that recent graph-based architectures are important references, and we explain below why several of them are not directly comparable or cannot be faithfully reproduced under our experimental constraints.
> > >
> > > (1) GraphCare[7] and RAM-EHR[8]: Both methods rely on large language models and knowledge-enhanced components to construct augmented representations. These approaches introduce substantial external knowledge resources far beyond the standard EHR-only setting of our study. Including them would confound the comparison by mixing knowledge-augmented and pure EHR sequence/graph models, making the evaluation unfair. Our goal is to benchmark DT-BEHRT strictly under EHR-only inputs (or with only minimal ontology structure information, such as HEART), consistent with prior Transformer and graph baselines.
> > >
> > > Although we do not incorporate substantial external knowledge in this work, our architecture is fully compatible with LLM-based augmentation. For example, large language models can be used to infer additional latent edges on the Disease Progression Graph, thereby enriching the relational structure and enabling more informative message passing without altering the core architecture.
> > >
> > > (2) GT-BEHRT: GT-BEHRT is indeed an important hybrid model; however, no complete, runnable implementation or training configuration is publicly available.

---

> > > > ### Author Response · Authors · 2025-12-03
> > > > **Response to Reviewer WRyC – W4**
> > > >
> > > > > **W4. All experiments focus on simple classifications (binary or multi-class). The absence of multi-label tasks such as drug recommendation or diagnosis prediction limits the assessment of the model’s scalability to more complex EHR applications.**
> > > >
> > > > We would like to clarify that DT-BEHRT has already been evaluated on a multi-label phenotype prediction task, as shown in Table 2. This task involves simultaneously predicting multiple phenotypes for each patient, and therefore directly measures the model’s ability to handle multi-label outputs beyond simple binary classification.

---

> > > > > ### Author Response · Authors · 2025-12-03
> > > > > **Response to Reviewer WRyC – W5**
> > > > >
> > > > > > **W5. While ablation results are provided, they do not test alternative pretraining schemes from prior work. Comparing the proposed objectives against those of G-BERT[2] or Sherbet[5] would more clearly justify the necessity and effectiveness of the new pretraining design.**
> > > > >
> > > > > We address it in two parts:
> > > > >
> > > > > (a) Pretraining schemes from Sherbet are not directly comparable due to fundamentally different training paradigms. Sherbet uses ontology-guided pretraining solely as an embedding initialization step. Its hierarchical objective is applied before model training and is not integrated into an end-to-end predictive framework. In contrast, DT-BEHRT learns hierarchical ancestor–children’s interactions dynamically and jointly with the downstream prediction task, enabling continuous task-specific refinement.
> > > > >
> > > > > (b) More fine-grained ablations of our pretraining tasks have been added. Specifically, we introduce two additional ablation variants for each proposed component, with the corresponding configurations alongside the base model provided below. We refer reviewers to Table 3 in the updated manuscript for the complete analysis. The results clearly delineate the individual and joint contributions of GCMP and ACP, directly demonstrating the necessity and effectiveness of the proposed pretraining design within DT-BEHRT.
> > > > >
> > > > > ### Table 3: Ablation study (partial).
> > > > > |**Dataset**    |    **Task**   | **Metric** |            |          |        |
> > > > > |-----------|--------------------|--------|---------------------|-----------------------|---------------------|
> > > > > |**Variant**| **Architectures**  | DA$^{w/o~cov}$ | ×                 | ✓                     | ×         |
> > > > > |           |                    | DA        | ×                 | ×                     | ×                   |
> > > > > |           |                    | DP        | ×                 | ×                     | ×                   |
> > > > > |           | **Pre-training Tasks** | GCMP  | ×                 | ×                     | ✓                   |
> > > > > |           |                    | ACP       | ×                 | ×                     | ×                   |
> > > > > | **MIMIC-III** | **Mortality**     | F1      | 71.59±1.29          | 72.01±1.76           | **74.17±0.09**          |
> > > > > |           |                    | AUROC   | 89.18±1.34          | 90.11±0.98            | **91.41±0.32**          |
> > > > > |           |                    | AUPRC   | 80.04±1.69          | 79.82±1.97            | **82.21±0.34**          |
> > > > > |           | **PLOS**               | F1      | 75.07±0.19          | 75.27±0.89            | **76.17±0.11**          |
> > > > > |           |                    | AUROC   | 81.67±0.56          | 81.95±0.96            | **83.51±0.44**          |
> > > > > |           |                    | AUPRC   | 82.43±0.47          | 82.30±1.21            | **84.22±0.43**          |
> > > > > |           | **Readmission**        | F1      | **70.39±0.32**          | 68.37±0.93            | 69.75±0.26          |
> > > > > |           |                    | AUROC   | 79.42±0.36          | 78.78±0.34            | **79.90±0.22**          |
> > > > > |           |                    | AUPRC   | 67.77±1.21          | 68.96±0.58            | **69.94±0.43**      |
> > > > > | **MIMIC-IV** | **Mortality**       | F1      | 63.84±2.09          | 65.70±1.13            | **67.78±0.70**          |
> > > > > |           |                    | AUROC  | 93.83±0.37          | 94.58±0.54            | **95.13±0.18**          |
> > > > > |           |                    | AUPRC   | 69.86±1.69          | 72.27±1.74            | **74.07±0.54**          |
> > > > > |           | **PLOS**               | F1      | 66.39±0.87          | **67.92±0.19**            | 67.86±0.26          |
> > > > > |           |                    | AUROC   | 83.72±0.38          | 84.67±0.18            | **84.77±0.06**    |
> > > > > |           |                    | AUPRC   | 73.33±0.68          | **75.29±0.16**        | 75.07±0.24    |
> > > > > |           | **Readmission**        | F1      | 83.79±0.11          | **83.82±0.08**            | 83.80±0.06          |
> > > > > |           |                    | AUROC   | 70.15±0.84          | **71.45±0.23**            | 70.94±0.36          |
> > > > > |           |                    | AUPRC   | 83.61±0.65          | **84.24±0.15**            | 83.86±0.28          |

---

> > > > > > ### Author Response · Authors · 2025-12-03
> > > > > > **Response to Reviewer WRyC – W6**
> > > > > >
> > > > > > > **W6. The model employs multi-head self-attention in the sequence encoder and graph attention propagation across both disease and visit nodes, which substantially increases computational overhead. Moreover, jointly learn multiple embeddings further raises concerns about the model’s practicality in clinical practices. The concern why most previous works avoid using both large-scale graphs and transformer together is related to efficiency problem, but DT-BEHRT combines both without discussing potential optimization or scalability issues.**
> > > > > >
> > > > > > We fully agree that combining multi-head self-attention with graph attention mechanisms increases computational cost, and we have explicitly discussed this as a limitation in the paper. We benchmarked the wall-clock runtime of one training epoch on the MIMIC-III dataset under identical batch size and optimization settings, using the default configurations for both G-BERT and Sherbet. DT-BEHRT requires ~12 s per epoch, compared with ~28 s for G-BERT and ~5 s for Sherbet, confirming that although DT-BEHRT introduces graph-enhanced transformer, the training overhead remains within acceptable offline training ranges. It is also worth noting that Sherbet attains its substantially lower computational cost largely because it models only diagnosis codes (based on the publicly released source code), while excluding other clinically informative codes such as medications, procedures, and laboratory results. This narrower feature space significantly reduces model complexity and contributes to its faster runtime.

---

> > > > > > > ### Author Response · Authors · 2025-12-03
> > > > > > > **Response to Reviewer WRyC – Reference**
> > > > > > >
> > > > > > > Reference:
> > > > > > >
> > > > > > > [1] Li, Yikuan, et al. "BEHRT: transformer for electronic health records." Scientific reports 10.1 (2020): 7155.
> > > > > > >
> > > > > > > [2] Rasmy, Laila, et al. "Med-BERT: pretrained contextualized embeddings on large-scale structured electronic health records for disease prediction." NPJ digital medicine 4.1 (2021): 86.
> > > > > > >
> > > > > > > [3] Choi, Edward, et al. "Learning the graphical structure of electronic health records with graph convolutional transformer." Proceedings of the AAAI conference on artificial intelligence. Vol. 34. No. 01. 2020.
> > > > > > >
> > > > > > > [4] Poulain, Raphael, and Rahmatollah Beheshti. "Graph transformers on EHRs: Better representation improves downstream performance." The Twelfth International Conference on Learning Representations. 2024.
> > > > > > >
> > > > > > > [5] Shang, Junyuan, et al. "Pre-training of graph augmented transformers for medication recommendation." arXiv preprint arXiv:1906.00346 (2019).
> > > > > > >
> > > > > > > [6] Lu, Chang, Chandan K. Reddy, and Yue Ning. "Self-supervised graph learning with hyperbolic embedding for temporal health event prediction." IEEE Transactions on Cybernetics 53.4 (2021): 2124-2136.
> > > > > > >
> > > > > > > [7] Jiang, Pengcheng, et al. "Graphcare: Enhancing healthcare predictions with personalized knowledge graphs." arXiv preprint arXiv:2305.12788 (2023).
> > > > > > >
> > > > > > > [8] Xu, Ran, et al. "Ram-ehr: Retrieval augmentation meets clinical predictions on electronic health records." Proceedings of the 62nd Annual Meeting of the Association for Computational Linguistics (Volume 2: Short Papers). 2024.

---

### Meta-Review · Area_Chair_cdby · 2026-01-01

**Summary:**

Reviewers agreed that the paper is well motivated and that DT-BEHRT’s modular design can yield competitive performance and qualitative case-study visualizations, but several concerns collectively lowered confidence in acceptance. The most significant issue was limited novelty: multiple reviewers viewed the architecture as an incremental combination of existing Transformer and graph design patterns, and noted that the proposed pretraining objectives (global masking and ancestor prediction) substantially overlap prior EHR pretraining/ontology-supervision approaches (e.g., G-BERT and related work), with insufficient justification that the new objectives are meaningfully different or necessary.

Reviewers also questioned experimental completeness and fairness, citing missing or outdated baselines (notably recent graph/graph-transformer models), limited evaluation diversity (task breadth and external validation), and the lack of controls that disentangle whether gains primarily come from pretraining versus the DA/DP modules. Finally, interpretability claims were viewed as overstated and not rigorously validated (largely relying on attention-style attributions and a small number of case studies), alongside concerns about unclear task definitions (e.g., readmission horizon) and practical efficiency/scalability when combining full self-attention with graph attention over large code sequences.

**Reviewer Concerns:**

The concerns on framework and experiments justification were addressed (at least partially) in the rebuttal, while the concerns on technical novelty/incremental contribution are still outstanding.

**Reviewer Scores:**

Reviewers who focused on experimental completeness, validation, and clarity (8c5g, CLPJ) would likely respond positively to the rebuttal, while those whose main objections centered on core novelty, interpretability rigor, or conceptual contribution (WRyC, MSyk, Y6Q5) would likely keep their original scores.

Overall, this paper would fall below the bar of acceptance.

---

### Decision · Program_Chairs · 2026-01-26

Reject